# ONLINE PRE-TRAINING FOR OFFLINE-TO-ONLINE REINFORCEMENT LEARNING

## ABSTRACT

Reinforcement Learning (RL) has achieved notable success in tasks requiring complex decision making, with offline RL offering the ability to train agents using fixed datasets, thereby avoiding the risks and costs associated with online interactions. However, offline RL is inherently limited by the quality of the dataset, which can restrict an agent's performance. Offline-to-online RL aims to bridge the gap between the cost-efficiency of offline RL and the performance potential of online RL by pre-training an agent offline before fine-tuning it through online interactions. Despite its promise, recent studies show that offline pre-trained agents often underperform during online fine-tuning due to inaccurate value function, with random initialization proving more effective in certain cases. In this work, we propose a novel method, Online Pre-Training for Offline-to-Online RL (OPT), to address the issue of inaccurate value estimation in offline pre-trained agents. OPT introduces a new learning phase, Online Pre-Training, which allows the training of a new value function that enhances the subsequent fine-tuning process. Implementation of OPT on TD3 and SPOT demonstrates an average 30% improvement in performance across D4RL environments, such as MuJoCo, Antmaze, and Adroit.

## 1 INTRODUCTION

Reinforcement Learning (RL) has shown great potential in addressing complex decision-making tasks across various fields (Mnih et al. 2015; Silver et al. 2017). In particular, offline RL (Levine et al. 2020) offers the advantage of training an agent on the fixed dataset, thereby mitigating the potential costs or risks associated with direct interactions in real-world environments - a significant limitation of online RL. However, the effectiveness of offline RL is inherently constrained by the quality of the dataset, which can impede the agent's overall performance.

To overcome the cost challenge of online RL and the performance limitation of offline RL, the offline-to-online RL approach has been introduced (Lee et al. 2022; Zhang et al. 2023; Yu & Zhang 2023). This approach entails training an agent sufficiently on an offline dataset, followed by fine-tuning through additional interactions with the environment. This allows the agent to utilize the knowledge acquired offline for online fine-tuning. Combining the strengths of both approaches, offline-to-online RL reduces the need for extensive environment interactions, while enhancing the agent's performance through online fine-tuning.

Although offline-to-online RL offers clear advantages, prior studies (Zhang et al. 2023; Guo et al. 2023; Nakamoto et al. 2024; Zhang et al. 2024; Kong et al. 2024; Hu et al. 2024) have shown that fine-tuning an offline pre-trained agent often results in worse performance compared to training from scratch. This phenomenon, described by Nakamoto et al. 2024 as *counter-intuitive trends*, is depicted in Figure 1, which compares the learning curves during the online phase for both fine-tuning and training from scratch with the replay buffer initialized using the offline dataset. As shown in Figure 1 (a), training from scratch outperforms fine-tuning from the very beginning of the learning process, as also observed by (Ball et al. 2023). In Figure 1 (b), although the offline pre-trained agent exhibits partial success, training from scratch eventually surpasses fine-tuning, highlighting the inherent challenges in fine-tuning pre-trained agents in offline settings.

Previous studies (Nakamoto et al. 2024; Zhang et al. 2024) attribute this *counter-intuitive trends* of online fine-tuning to issues stemming from inaccurate value estimation. In response, Nakamoto et al. 2024 focuses on providing a lower bound for value updates to correct the value estimation.

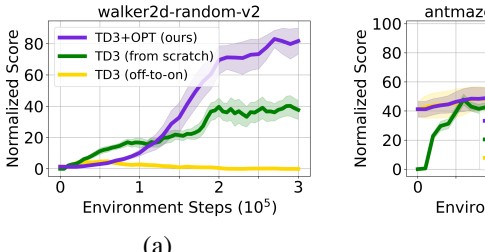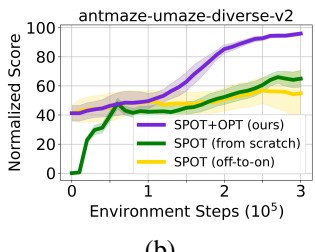

(a)                    (b)

Figure 1: Comparison between offline-to-online RL (yellow), from scratch (green), and our method (purple). (a): A TD3+BC (Fujimoto & Gu 2021) pre-trained agent is fine-tuned with TD3 (Fujimoto et al. 2018). (b): A SPOT pre-trained agent is fine-tuned with the same algorithm.

Meanwhile, Zhang et al. 2024 introduces perturbations in value updates, promoting smoother value estimation. A notable characteristic of these approaches is their reliance on the same value function across both the offline and online learning phases. However, we approach this issue from a different angle. Rather than relying solely on the potentially flawed value function, we propose introducing and utilizing an entirely new value function. To investigate this, we formulate two key research questions that guide our exploration:

Q1. *"Can adding a new value function resolve the issue of slow performance improvement?"*

Q2. *"How can we best leverage the new value function during online fine-tuning?"*

Through a comprehensive analysis of these research questions, we propose Online Pre-Training for Offline-to-Online RL (OPT), a novel approach that introduces a new value function to leverage it during online fine-tuning. In response to the first research question, OPT introduces a new value function, enhancing overall performance, as illustrated in Figure 1. For the second research question, OPT incorporates an additional learning phase, termed Online Pre-Training, which focuses on learning this new value function.

As OPT involves learning by adding a new value function, it can be broadly applied to value function-based RL methods. We evaluate OPT across various D4RL environments, including MuJoCo, Antmaze, and Adroit. OPT demonstrates an average 30% improvement within a limited setting of 300k online interactions in the final normalized score, surpassing previous state-of-the-art results.

## 2   Background and Related Work

Reinforcement Learning (RL) is modeled as a Markov Decision Process (MDP) (Puterman 1990). In this framework, at each time step, an agent selects an action $a$ based on its current state $s$ according to its policy $\pi(a|s)$. The environment transitions to a subsequent state $s'$ and provides a reward $r$, following the transition probability $p(s'|s, a)$ and reward function $r(s, a)$, respectively. Over successive interactions, the agent's policy $\pi$ is optimized to maximize the expected cumulative return $\mathbb{E}_\pi \left[ \sum_t \gamma^t r(s_t, a_t) \right]$, where $\gamma \in [0, 1)$ is the discount factor.

Offline RL focuses on training agents using a static dataset $D = \{(s, a, r, s')\}$, usually generated by various policies. To address the constraints of offline RL, which are often limited by the dataset's quality, offline-to-online RL introduces an additional phase of online fine-tuning. This hybrid approach enhances the agent's performance by allowing further learning directly from interactions with the environment.

**Addressing Inaccurate Offline Value Estimation.**   Offline RL, reliant on a fixed dataset, is prone to extrapolation error when the value function evaluates out-of-distribution (OOD) actions (Kumar et al. 2020; Fujimoto et al. 2019). Several methods have been proposed to address this challenge: some focus on training the value function to assign lower values to OOD actions (Wu et al. 2019; Kostrikov et al. 2021a), while others aim to avoid OOD action evaluation altogether (Kostrikov et al. 2021b). These inaccurate value estimations in offline training not only degrade offline performance

but can also adversely impact subsequent online fine-tuning (Yu & Zhang 2023; Zhang et al. 2024; Nakamoto et al. 2024; Kong et al. 2024; Feng et al. 2024). Although constraints applied in offline RL can be extended to online fine-tuning to mitigate this issue (Kostrikov et al. 2021b; Lyu et al. 2022; Wu et al. 2022), such strategies often impose excessive conservatism, limiting the potential for performance enhancement.

Recent advances in offline-to-online RL have aimed to overcome this limitation arising from inaccurate value estimations (Nakamoto et al. 2024; Zhang et al. 2024). One approach (Nakamoto et al. 2024) addresses over-conservatism during the offline phase by providing a lower bound for value updates. However, the conservative nature of this method continues to hinder policy improvement. Another approach (Zhang et al. 2024) introduces perturbations to value updates and increases their update frequency. While effective, this method incurs significantly higher computational costs compared to standard techniques, making it less practical for general use.

**Backbone Algorithms.** Our proposed Online Pre-Training process using newly introduced value function can be applied to various backbone algorithms. Among the many potential candidates, this study utilizes TD3+BC (Fujimoto & Gu 2021) and SPOT (Wu et al. 2022) as the backbone algorithms due to their simplicity and sample efficiency. TD3+BC (Fujimoto & Gu 2021) extends the original TD3 (Fujimoto et al. 2018) algorithm by incorporating a behavior cloning (BC) regularization term into the policy improvement. The value function is trained using temporal-difference (TD) learning, with the loss functions for both the policy $\pi_\phi(s)$ and value function $Q_\theta(s, a)$ defined as follows:

$$\mathcal{L}_\pi(\phi) = \mathbb{E}_{s \sim B}[-Q_\theta(s, \pi_\phi(s)) + \alpha(\pi_\phi(s) - a)^2], \tag{1}$$

$$\mathcal{L}_Q(\theta) = \mathbb{E}_{(s,a,r,s') \sim B}[(Q_\theta(s, a) - (r + \gamma Q_{\bar{\theta}}(s', \pi_\phi(s'))))^2] \tag{2}$$

where $Q_{\bar{\theta}}$ denotes a delayed target value function and $B$ is the replay buffer. SPOT (Wu et al. 2022) extends of TD3+BC by replacing the BC regularization term with a pre-trained VAE, which is then used to penalize OOD actions based on the uncertainty.

## 3 METHOD

In this section, we introduce Online Pre-Training for Offline-to-Online RL (OPT), a novel method aimed at addressing the issue of inaccurate value estimation in offline-to-online RL. The proposed method employs two value functions, $Q^{\text{off-pt}}$ and $Q^{\text{on-pt}}$, each serving distinct roles in the time domain. The method consists of three distinct stages of learning:

(i) Offline Pre-Training: As in conventional offline-to-online RL, $Q^{\text{off-pt}}$ is trained on the offline dataset, yielding the offline pre-trained policy $\pi^{\text{off}}$.

(ii) Online Pre-Training: The second value function, $Q^{\text{on-pt}}$, is trained using both the offline dataset and newly collected online samples.

(iii) Online Fine-Tuning: The policy is updated by utilizing both $Q^{\text{off-pt}}$ and $Q^{\text{on-pt}}$, with each value function continuously updated.

Figure 2 illustrates the difference between conventional offline-to-online learning (Figure 2a) and our proposed method, OPT (Figure 2b). OPT introduces a new learning phase called Online Pre-Training, making it distinct from the conventional two-stage offline-to-online RL methods by comprising three stages. The following sections focus on the Online Pre-Training and Online Fine-Tuning phases, while the offline phase adheres to the standard offline RL process.

### 3.1 ONLINE PRE-TRAINING

In the proposed method, $Q^{\text{on-pt}}$ is introduced as an additional value function specifically designed for online fine-tuning. One straightforward approach is to add a randomly initialized value function. As $Q^{\text{on-pt}}$ begins learning from the online fine-tuning, it is expected to adapt well to the new data encountered during online fine-tuning. However, since $Q^{\text{on-pt}}$ is required to train from scratch, it often disrupts policy learning in the early stages. To prevent $Q^{\text{on-pt}}$ from disrupting policy learning, we introduce a pre-training phase, termed Online Pre-Training, specifically designed to train $Q^{\text{on-pt}}$. The following sections explore the design of the Online Pre-Training method in detail.

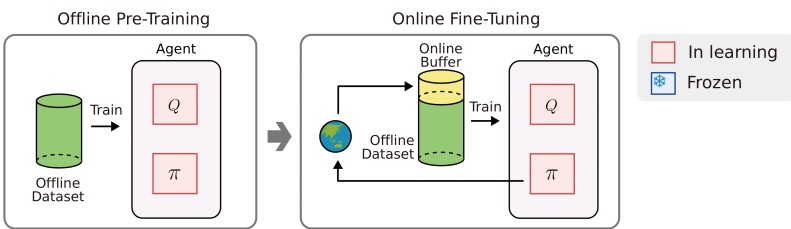

(a) Conventional Offline-to-Online RL

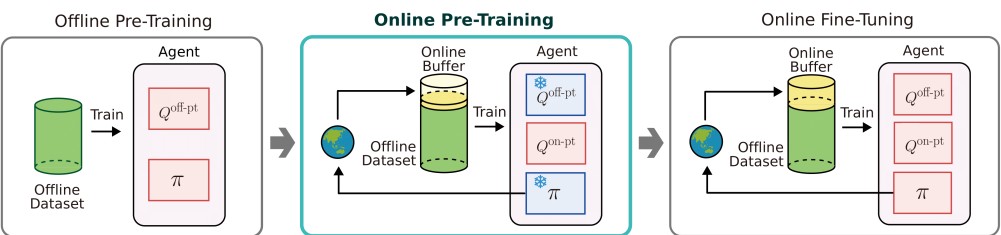

(b) **Ours:** Online Pre-Training for Offline-to-Online RL

Figure 2: Illustrations of two different learning methods: (a) Conventional Offline-to-Online RL (b) **Ours**. OPT introduces a new learning phase, termed Online Pre-Training, between offline pre-training and online fine-tuning. The illustration indicates whether the value function and policy are *in learning* or *frozen* (not being trained) during each training phase.

**Designing Datasets.**    As the initial stage of Online Pre-Training, the only available dataset is the offline dataset $B_{\text{off}}$, which has already been fully leveraged to train $Q^{\text{off-pt}}$. Relying solely on $B_{\text{off}}$ causes $Q^{\text{on-pt}}$ to closely replicate $Q^{\text{off-pt}}$. To address this, we incorporate online samples by initiating Online Pre-Training with the collection of $N_\tau$ samples in the online buffer $B_{\text{on}}$, generated by $\pi^{\text{off}}$. To leverage $B_{\text{on}}$, one approach is to train $Q^{\text{on-pt}}$ with $B_{\text{on}}$. Since $Q^{\text{on-pt}}$ is trained based on $\pi^{\text{off}}$, this prevents $Q^{\text{on-pt}}$ from disrupting policy learning in the initial stage of online fine-tuning. However, as $B_{\text{on}}$ is generated by the fixed policy $\pi^{\text{off}}$, relying solely on $B_{\text{on}}$ risks overfitting. Therefore, to address both issue of similarity to $Q^{\text{off-pt}}$ and the risk of overfitting to $\pi^{\text{off}}$, a balanced approach utilizing both $B_{\text{off}}$ and $B_{\text{on}}$ is necessary during training.

**Designing Objective Function.**    By leveraging both datasets, the objective for $Q^{\text{on-pt}}$ is to ensure its adaptability to the evolving policy samples, promoting continuous policy improvement and enhancing sample efficiency. To achieve this, we adopt a meta-adaptation strategy based on OEMA (Guo et al. 2023). The objective function for $Q^{\text{on-pt}}$ is outlined as follows:

$$\mathcal{L}^{\text{pretrain}}_{Q^{\text{on-pt}}}(\psi) = \mathcal{L}^{\text{off}}_{Q^{\text{on-pt}}}(\psi) + \mathcal{L}^{\text{on}}_{Q^{\text{on-pt}}}(\psi - \alpha\nabla\mathcal{L}^{\text{off}}_{Q^{\text{on-pt}}}(\psi)) \tag{3}$$

$$\text{where}\quad \mathcal{L}^{\text{off}}_{Q^{\text{on-pt}}}(\psi) = \mathbb{E}_{s,a,r,s'\in B_{\text{off}}}[(Q^{\text{on-pt}}_{\psi}(s,a) - (r + \gamma Q^{\text{on-pt}}_{\bar{\psi}}(s',\pi_\phi(s'))))^2],$$

$$\mathcal{L}^{\text{on}}_{Q^{\text{on-pt}}}(\psi) = \mathbb{E}_{s,a,r,s'\in B_{\text{on}}}[(Q^{\text{on-pt}}_{\psi}(s,a) - (r + \gamma Q^{\text{on-pt}}_{\bar{\psi}}(s',\pi_\phi(s'))))^2].$$

Here, $Q^{\text{on-pt}}_{\bar{\psi}}$ represents the target network. Equation (3) consists of two terms: the first term facilitates learning from $B_{\text{off}}$, while the second term serves as an objective to ensure that $Q^{\text{on-pt}}$ adapts to $B_{\text{on}}$. Optimizing these terms allows $Q^{\text{on-pt}}$ to leverage $B_{\text{off}}$ and align closely with the dynamics of the current policy, enabling efficient adaptation to online samples during fine-tuning. During the Online Pre-Training, only $Q^{\text{on-pt}}$ is updated, with no alterations made to other components, $\pi^{\text{off}}$ and $Q^{\text{off-pt}}$.

---

**Algorithm 1** OPT: Online Pre-Training for Offline-to-Online Reinforcement Learning

---

1: **Inputs:** Offline dataset $\mathcal{B}_{\text{off}}$, offline trained agent $\{Q_\theta^{\text{off-pt}}, \pi_\phi\}$, online pre-training samples $N_\tau$, online pre-training steps $N_{\text{pretrain}}$, online fine-tuning steps $N_{\text{finetune}}$, weighting coefficient $\kappa$
2: Initialize online replay buffer $\mathcal{B}_{\text{on}}$, value function $Q_\psi^{\text{on-pt}}$
   `// Online Pre-Training`
3: Store $N_\tau$ transitions $\tau = (s, a, r, s')$ in $\mathcal{B}_{\text{on}}$ via environment interaction with $\pi_\phi$
4: **for** $i = 1$ **to** $N_{\text{pretrain}}$ **do**
5:     Sample minibatch of transitions $\{\tau_j\}_{j=1}^B \sim \mathcal{B}_{\text{off}}, \{\tau_j\}_{j=1}^B \sim \mathcal{B}_{\text{on}}$
6:     Update $\psi$ minimizing $\mathcal{L}_{Q^{\text{on-pt}}}^{\text{pretrain}}(\psi)$ by Equation 3
7: **end for**
   `// Online Fine-Tuning`
8: Initialize balanced replay buffer $\mathcal{B}_{\text{BR}} \leftarrow \mathcal{B}_{\text{off}} \cup \mathcal{B}_{\text{on}}$
9: **for** $i = 1$ **to** $N_{\text{finetune}}$ **do**
10:     Sample minibatch of transitions $\tau \sim \mathcal{B}_{\text{BR}}$
11:     Update $\theta$ and $\psi$ minimizing $\mathcal{L}_{Q^{\text{off-pt}}}(\theta)$ and $\mathcal{L}_{Q^{\text{on-pt}}}(\psi)$ respectively by Equation 2
12:     Update $\phi$ minimizing $\mathcal{L}_\pi^{\text{finetune}}(\phi)$ by Equation 4
13: **end for**

---

## 3.2 ONLINE FINE-TUNING

In online fine-tuning, by utilizing $Q^{\text{on-pt}}$ trained during Online Pre-Training, we facilitate effective policy improvement. Throughout this phase, the buffer $B_{\text{on}}$ is continuously filled and the learning of all three components $Q^{\text{off-pt}}$, $Q^{\text{on-pt}}$ and $\pi_\phi$ progresses. As in the conventional offline-to-online RL, $Q^{\text{off-pt}}$ is updated using TD learning, and similarly, $Q^{\text{on-pt}}$, which was trained by Equation 3, is also updated via TD learning.

**Effectively Balancing $Q^{\text{off-pt}}$ and $Q^{\text{on-pt}}$.** One of the key aspects of OPT lies in its approach to policy improvement, which effectively balances $Q^{\text{off-pt}}$ and $Q^{\text{on-pt}}$ during online fine-tuning. While most previous works rely solely on a single value function, our method leverages both $Q^{\text{off-pt}}$ and $Q^{\text{on-pt}}$ for policy improvement. Since $Q^{\text{off-pt}}$ is informative for the offline dataset and $Q^{\text{on-pt}}$, pre-trained via meta-adaptation strategy during Online Pre-Training, can be adapted to new data encountered through online interactions, effectively leveraging both value functions is central to our online fine-tuning strategy. The proposed loss function for policy improvement during online fine-tuning is given by:

$$\mathcal{L}_\pi^{\text{finetune}}(\phi) = \mathbb{E}_{s \sim B}[-\{(1 - \kappa)Q^{\text{off-pt}}(s, \pi_\phi(s)) + \kappa Q^{\text{on-pt}}(s, \pi_\phi(s))\}], \tag{4}$$

where $\pi_\phi$ is initialized as $\pi^{\text{off}}$ and $0 < \kappa \leq 1$. $\kappa$ is a weighting coefficient that balances the ratio between $Q^{\text{off-pt}}$ and $Q^{\text{on-pt}}$. When the discrepancy between the offline dataset and newly introduced online samples is minimal, $Q^{\text{off-pt}}$ retains valuable information from the offline dataset. As a result, a small $\kappa$ is employed to more effectively leverage $Q^{\text{off-pt}}$. As online fine-tuning progresses, $Q^{\text{on-pt}}$, optimized through our meta-adaptation objective, quickly adapts to the online data. Consequently, $\kappa$ is incrementally increased to shift the reliance towards the more rapidly adapting $Q^{\text{on-pt}}$. A detailed analysis of $\kappa$ is provided in Section 5.2.

Additionally, to promote the use of online samples, we employ balanced replay (Lee et al. 2022), which prioritizes samples encountered during online interactions to further accelerate the adaptation of $Q^{\text{on-pt}}$. The overall learning phase of OPT is illustrated in Figure 2b, with the algorithm presented in Algorithm 1.

## 4 EXPERIMENTS

In this section, we demonstrate the effectiveness of our proposed method through experimental results. In Section 4.1, we describe our experimental setup and compare our method against existing offline-to-online RL approaches across various environments. Section 4.2 explores the application of our method to an alternative backbone algorithm.

## 4.1 MAIN RESULTS

**Experimental Setup.** We evaluate the performance of OPT across three domains from the D4RL benchmark (Fu et al. 2020). MuJoCo is a suite of locomotion tasks including datasets of diverse quality for each environment. Antmaze is a set of navigation tasks where an ant robot is controlled to navigate from a starting point to a goal location within a maze. Adroit is a set of robot manipulation tasks that require controlling a five-finger robotic hand to achieve a specific goal in each task. A detailed description of the environment and dataset is provided in Appendix B. For all baselines, the offline phase comprises 1M gradient steps, and the online phase consists of 300k environment steps. OPT carries out Online Pre-Training for the first 25k steps of the online phase, followed by online fine-tuning for the remaining 275k steps thus, like the other baselines, it also has a total 300k environment steps in the online phase. The implementation details are provided in Appendix A.1.

**Baselines.** We compare OPT with the following baselines: (1) Off2On (Lee et al. 2022), an ensemble-based method that incorporates balanced replay to promote the use of near-on-policy samples; (2) OEMA (Guo et al. 2023), which applies an optimistic strategy alongside a meta-adaptation method for policy learning; (3) PEX (Zhang et al. 2023), which utilizes a set of policies, including a frozen offline pre-trained policy and an additional learnable policy; (4) ACA (Yu & Zhang 2023), which post-process the offline pre-trained value function to align it with the policy; (5) FamO2O (Wang et al. 2024), employing a state-adaptive policy improvement method; and (6) Cal-QL (Nakamoto et al. 2024), which trains a value function to mitigate over-conservatism introduced during the offline pre-training.

Table 1: Comparison of the normalized scores after online fine-tuning for each environment in MuJoCo domain. r = random, m = medium, m-r = medium-replay. All results are reported as the mean and standard deviation across five random seeds.

| Environment | TD3 | Off2On | OEMA | PEX | ACA | FamO2O | Cal-QL | TD3 + OPT (Ours) |
|---|---|---|---|---|---|---|---|---|
| halfcheetah-r | **96.9**$_{\pm4.9}$ | 92.6$_{\pm5.6}$ | 78.9$_{\pm13.0}$ | 60.9$_{\pm5.0}$ | 92.0$_{\pm2.5}$ | 36.9$_{\pm3.5}$ | 32.8$_{\pm8.0}$ | 89.0$_{\pm2.1}$ |
| hopper-r | 84.4$_{\pm30.1}$ | 95.3$_{\pm9.1}$ | 49.1$_{\pm28.2}$ | 48.5$_{\pm38.9}$ | 81.1$_{\pm27.2}$ | 11.8$_{\pm2.0}$ | 17.7$_{\pm26.0}$ | **109.5**$_{\pm3.1}$ |
| walker2d-r | 0.1$_{\pm0.0}$ | 27.9$_{\pm2.2}$ | 24.5$_{\pm22.7}$ | 9.8$_{\pm1.6}$ | 33.8$_{\pm23.0}$ | 9.3$_{\pm0.3}$ | 9.3$_{\pm5.6}$ | **88.1**$_{\pm5.2}$ |
| halfcheetah-m | 96.1$_{\pm1.8}$ | **103.3**$_{\pm1.5}$ | 58.5$_{\pm33.0}$ | 70.4$_{\pm2.3}$ | 80.6$_{\pm1.0}$ | 49.6$_{\pm0.3}$ | 76.9$_{\pm2.1}$ | 96.6$_{\pm1.7}$ |
| hopper-m | 84.5$_{\pm30.3}$ | 106.3$_{\pm1.2}$ | 107.7$_{\pm2.8}$ | 86.1$_{\pm26.3}$ | 102.8$_{\pm0.5}$ | 77.7$_{\pm7.8}$ | 100.6$_{\pm0.8}$ | **112.0**$_{\pm1.3}$ |
| walker2d-m | 102.0$_{\pm8.0}$ | 109.7$_{\pm29.6}$ | 92.2$_{\pm8.7}$ | 91.4$_{\pm14.3}$ | 87.1$_{\pm3.4}$ | 83.7$_{\pm2.5}$ | 97.0$_{\pm8.2}$ | **116.1**$_{\pm4.7}$ |
| halfcheetah-m-r | 87.5$_{\pm1.5}$ | **95.6**$_{\pm1.6}$ | 30.8$_{\pm27.6}$ | 55.4$_{\pm1.2}$ | 66.2$_{\pm2.8}$ | 48.3$_{\pm0.6}$ | 62.1$_{\pm1.0}$ | 92.2$_{\pm1.2}$ |
| hopper-m-r | 90.9$_{\pm25.4}$ | 101.6$_{\pm14.8}$ | 108.8$_{\pm1.8}$ | 95.3$_{\pm7.2}$ | 105.8$_{\pm0.9}$ | 102.1$_{\pm0.7}$ | 101.4$_{\pm2.1}$ | **112.7**$_{\pm1.1}$ |
| walker2d-m-r | 107.7$_{\pm7.4}$ | **120.2**$_{\pm9.3}$ | 103.9$_{\pm5.3}$ | 87.2$_{\pm13.6}$ | 79.5$_{\pm30.1}$ | 91.3$_{\pm6.9}$ | 98.3$_{\pm3.2}$ | 117.7$_{\pm3.5}$ |
| Total | 747.2 | 852.5 | 654.4 | 605.0 | 728.9 | 510.1 | 596.1 | **933.9** |

**Results on MuJoCo.** In the MuJoCo, we implement OPT on the baseline which utilizes TD3+BC for the offline phase, followed by TD3 in the online phase. Both TD3+OPT and TD3 are evaluated using an update-to-data (UTD) ratio of 5 for consistency. The results in Table 1 indicate that OPT demonstrates strong overall performance, notably surpassing the existing state-of-the-art (SOTA) in several environments. The comparatively high total score further highlights OPT's robustness, illustrating its capability to perform consistently well across a range of environments and datasets. In particular, the results for the `walker2d-random-v2` dataset demonstrate the remarkable efficacy of OPT, as it significantly outperforms existing approaches.

**Results on Antmaze.** In the Antmaze, we implement OPT within the SPOT, as TD3 showed suboptimal performance in this domain. Table 2 shows that OPT consistently delivers superior performance across all environments. When comparing to Cal-QL (Nakamoto et al. 2024), a recently proposed method with the same objective of addressing inaccurate value estimation, the results in the `umaze-diverse` and `large-diverse` environments demonstrate the efficacy of introducing a new value function through Online Pre-Training in mitigating this issue.

**Results on Adroit.** In the Adroit, as with Antmaze, we apply OPT to the SPOT. Table 3 demonstrates that, unlike Cal-QL (Nakamoto et al. 2024), which struggles to learn from low-quality datasets such as `cloned` due to its conservative nature, OPT manages to perform well even with

Table 2: Comparison of the normalized scores after online fine-tuning for each environment in Antmaze domain. All results are reported as the mean and standard deviation across five random seeds.

| Environment | SPOT | PEX | ACA | FamO2O | Cal-QL | SPOT + OPT (Ours) |
|---|---|---|---|---|---|---|
| umaze | $98.4_{\pm1.8}$ | $95.2_{\pm1.6}$ | $92.0_{\pm4.6}$ | $94.6_{\pm2.0}$ | $90.1_{\pm10.8}$ | $\mathbf{99.8}_{\pm0.4}$ |
| umaze-diverse | $55.2_{\pm32.7}$ | $34.8_{\pm30.0}$ | $92.0_{\pm7.8}$ | $39.8_{\pm23.2}$ | $75.2_{\pm35.0}$ | $\mathbf{97.4}_{\pm0.4}$ |
| medium-play | $91.2_{\pm3.8}$ | $83.4_{\pm2.3}$ | $0.0_{\pm0.0}$ | $88.0_{\pm2.2}$ | $95.1_{\pm6.3}$ | $\mathbf{98.2}_{\pm1.3}$ |
| medium-diverse | $91.6_{\pm3.4}$ | $86.6_{\pm4.9}$ | $0.0_{\pm0.0}$ | $69.0_{\pm31.8}$ | $96.3_{\pm4.8}$ | $\mathbf{98.4}_{\pm1.3}$ |
| large-play | $60.4_{\pm21.4}$ | $56.0_{\pm3.8}$ | $0.0_{\pm0.0}$ | $53.8_{\pm7.6}$ | $75.0_{\pm14.7}$ | $\mathbf{78.2}_{\pm4.4}$ |
| large-diverse | $69.4_{\pm23.7}$ | $60.4_{\pm6.8}$ | $0.0_{\pm0.0}$ | $53.6_{\pm7.2}$ | $74.4_{\pm11.8}$ | $\mathbf{90.6}_{\pm3.7}$ |
| Total | 466.2 | 416.4 | 184.0 | 398.8 | 506.1 | **562.6** |

Table 3: Comparison of the normalized scores after online fine-tuning for each environment in Adroit domain. All results are reported as the mean and standard deviation across five random seeds.

| Environment | SPOT | Cal-QL | SPOT + OPT (Ours) |
|---|---|---|---|
| pen-cloned | $117.1_{\pm13.4}$ | $-2.0_{\pm1.2}$ | $\mathbf{130.3}_{\pm6.8}$ |
| hammer-cloned | $90.1_{\pm23.2}$ | $0.21_{\pm0.07}$ | $\mathbf{120.1}_{\pm4.2}$ |
| door-cloned | $0.04_{\pm0.06}$ | $-0.03_{\pm0.00}$ | $\mathbf{50.4}_{\pm29.2}$ |
| relocate-cloned | $-0.19_{\pm0.04}$ | $-0.33_{\pm0.01}$ | $\mathbf{-0.11}_{\pm0.06}$ |
| Total | 207.0 | -2.1 | 300.6 |

these challenging datasets. In particular, the results in the `door-cloned` environment, where SPOT fails to perform adequately, demonstrate the effectiveness of OPT.

## 4.2 EXTENDING OPT TO IQL

So far, we have shown that OPT is an effective algorithm for online fine-tuning when applied to TD3-based algorithms, such as TD3 and SPOT. To further validate the versatility of OPT across different baselines, we changed the baseline to IQL. Unlike TD3, IQL employs a stochastic policy and uses both a state-action value function and a state value function. Accordingly, to implement OPT, we modified the training loss, with detailed explanations provided in Appendix A.2.

The results in Figure 3 show that applying OPT yields noticeable performance improvement compared to the baseline. This suggests that OPT can be applied across various baseline algorithms. Additional experiment results in different domains, including MuJoCo and Adroit, are presented in Appendix E.

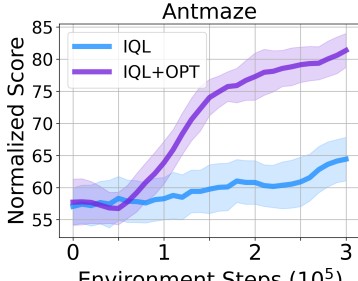

Figure 3: Aggregated return curves for IQL and IQL+OPT, averaged across all six environments in the Antmaze domain.

## 5 DISCUSSION

### 5.1 COMPARISON OPT WITH DIFFERENT INITIALIZATION METHODS

In Section 3.1, we explore various approaches for initializing $Q^{\text{on-pt}}$. To further substantiate the effectiveness of our initialization method through experimental results, we evaluate three approaches for initializing $Q^{\text{on-pt}}$: (1) Random Initialization, (2) Pre-trained with $B_{\text{on}}$, and (3) our proposed method, OPT.

Figure 4 illustrates that random initialization consistently underperforms compared to OPT. This discrepancy, as discussed in Section 3.1, can be attributed to the adverse impact of random initialization, which hinders policy learning and consequently leads to diminished performance. Similarly, the results for pre-training with $B_{\mathrm{on}}$ demonstrate that learning solely from $B_{\mathrm{on}}$ is insufficient to follow the performance of OPT. In particular, the results on the random dataset, where policy evolves more drastically as shown in Figure 5, the results demonstrate that an overfitted $Q^{\mathrm{on\text{-}pt}}$ fails to learn with this policy improvement. This underscores the importance of both $B_{\mathrm{off}}$ and $B_{\mathrm{on}}$ for training. Detailed experiment results are provided in Appendix C.3.

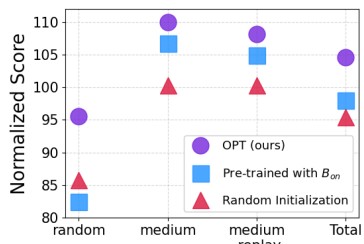

Figure 4: Comparing normalized score of OPT with different initialization methods, averaging across all 3 environments in the MuJoCo domain.

## 5.2 Correlation between Weighting Coefficient ($\kappa$) and Dataset

In our proposed method, we utilize $\kappa$ to assign the weights between $Q^{\mathrm{off\text{-}pt}}$ and $Q^{\mathrm{on\text{-}pt}}$ during online fine-tuning. Since the quality of the dataset affects the overall learning process, we adjust the $\kappa$ scheduling accordingly. To better understand the scheduling approach, we visualize the distributional differences between the dataset and the policy rollouts and examine their association with $\kappa$ scheduling. To this end, we compare the state-action distributions of the offline dataset and the samples generated by the policy using t-SNE (Van der Maaten & Hinton 2008) in the walker2d environment.

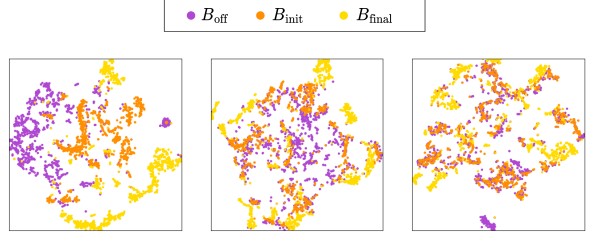

(a) Random    (b) Medium-Replay    (c) Medium

Figure 5: A t-SNE visualization of the offline dataset ($B_{\mathrm{off}}$) and the policy's rollout samples at the beginning ($B_{\mathrm{init}}$) and end ($B_{\mathrm{final}}$) of the online fine-tuning.

Figure 5 shows the comparison of the distribution between the offline dataset ($B_{\mathrm{off}}$) and the rollout samples of policy at both the beginning ($B_{\mathrm{init}}$) and the end ($B_{\mathrm{final}}$) of the online fine-tuning phase. For the medium and medium-replay datasets, we observe that the distributions are similar at the start of the online fine-tuning but diverse towards the end. In contrast, for the random dataset, a difference between the two distributions is evident from the beginning. Since $Q^{\mathrm{off\text{-}pt}}$ is informative for the offline dataset due to its training during offline pre-training, we initially assign it a higher weight during the early stages of online fine-tuning for the medium and medium-replay datasets. As the online fine-tuning progresses, the weight is gradually shifted toward $Q^{\mathrm{on\text{-}pt}}$. However, for the random dataset, due to the substantial distribution difference, we primarily rely on $Q^{\mathrm{on\text{-}pt}}$ from the start of online fine-tuning. We provide the specific values of $\kappa$ in Appendix D and present an ablation study in Appendix C.2.

## 5.3 Impact of Addition of a New Value Function

In our proposed algorithm, we introduce a new value function, which is trained during the Online Pre-Training and subsequently utilized in the online fine-tuning. To evaluate the impact of this addition, we examine the performance when the new value function is excluded. Specifically, we assess the outcome where $Q^{\mathrm{off\text{-}pt}}$ is trained during Online Pre-Training, without introducing $Q^{\mathrm{on\text{-}pt}}$ (denoted as w/o $Q^{\mathrm{on\text{-}pt}}$ in Figure 6). Under this setup, policy improvement in online fine-tuning is driven solely by $Q^{\mathrm{off\text{-}pt}}$.

The results presented in Figure 6 indicate that the addition of a new value function leads to improved performance regardless of the dataset. Notably, this improvement is pronounced in the

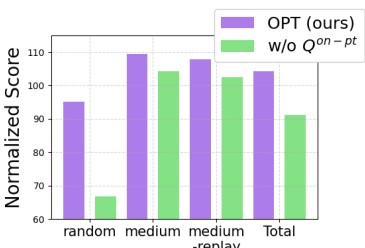

Figure 6: Comparison of the normalized score for OPT and its ablation (without the addition of a new value function), averaged across all 3 environments in the MuJoCo domain.

random dataset. Due to the characteristics of the random dataset, where successful demonstrations are limited, $Q^{\text{off-pt}}$, trained extensively on this dataset, becomes significantly biased and fails to benefit from Online Pre-Training.

## 5.4 What is the Effect of Online Pre-Training Samples ($N_\tau$)?

The Online Pre-Training phase involves two hyperparameters: $N_\tau$ and $N_{\text{pretrain}}$. In particular, $N_\tau$ represents the number of interactions with the environment used to collect online samples for Online Pre-Training. Since $N_\tau$ is also part of the environment steps within the online phase, determining the most efficient value for $N_\tau$ is critical for optimizing sample efficiency in offline-to-online RL. To investigate this, we conduct experiments in the MuJoCo domain comparing the results when $N_\tau$ is set to 1/4, 1/2, 2, and 4 times its current value (25000). In these experiments, $N_{\text{pretrain}}$ is set to twice the value of $N_\tau$, while the number of online fine-tuning steps is kept constant.

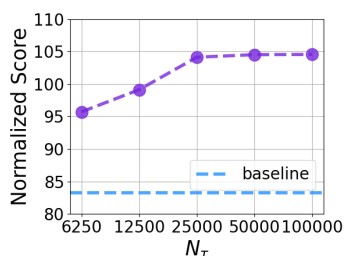

Figure 7: Comparing normalized score with varying $N_\tau$ of OPT, averaged across all 9 environments (3 tasks with 3 datasets each) in the MuJoCo doamin

The results in Figure 7 demonstrate that increasing $N_\tau$ enhances the effectiveness of Online Pre-Training. However, beyond a certain threshold, further increases in $N_\tau$ do not lead to additional performance gains. This is because during the environment interactions for $N_\tau$, the policy remained fixed, and once the amount of online data surpasses an optimal level, it no longer contributes to Online Pre-Training. Since the primary objective of offline-to-online RL is to achieve high performance with minimal environment interaction, these results suggest that OPT is most efficient when $N_\tau$ is set to 25000. Detailed results for each environment are presented in Appendix C.4.

## 5.5 Comparison with RLPD

Table 4: Normalized score for each environment on the MuJoCo domain. The full results for other domains, including Antmaze and Adroit, are provided in Appendix F.

| Environment | RLPD | | |
|---|---|---|---|
| | Vanilla | Off-to-On | OPT (Ours) |
| halfcheetah-r | 91.5 ±2.5 | 96.1 ±5.2 | 90.7 ±2.2 |
| hopper-r | 90.2 ±19.1 | 95.7 ±18.4 | **103.5** ±3.6 |
| walker2d-r | 87.7 ±14.1 | 74.3 ±13.9 | 79.2 ±10.0 |
| halfcheetah-m | 95.5 ±1.5 | 96.6 ±0.9 | 96.7 ±1.4 |
| hopper-m | 91.4 ±27.8 | 93.6 ±13.9 | **106.9** ±1.5 |
| walker2d-m | 121.6 ±2.3 | 124.1 ±2.4 | 122.8 ±3.0 |
| halfcheetah-m-r | 90.1 ±1.3 | 90.0 ±1.4 | 91.6 ±2.1 |
| hopper-m-r | 78.9 ±24.5 | 94.7 ±26.8 | **107.4** ±1.9 |
| walker2d-m-r | 119.0 ±2.2 | 122.5 ±2.7 | 120.9 ±2.3 |
| MuJoCo total | 866.0 | 887.6 | **918.7** |

Thus far, we proposed a method to integrate offline pre-trained agents into online fine-tuning effectively. Recently, several studies have emerged demonstrating strong performance using online RL alone with offline datasets, without requiring an explicit offline phase (Song et al. 2022; Ball et al. 2023). Among these, RLPD (Ball et al. 2023) has demonstrated state-of-the-art performances through the use of ensemble techniques and a high UTD (update-to-data) ratio. To assess the performance of integrating OPT with RLPD, we extend RLPD by incorporating an offline phase, followed by online fine-tuning, and subsequently apply OPT to evaluate its effectiveness. Further implementation details are provided in Appendix A.3.

Table 4 reports the results of original RLPD ('Vanilla'), RLPD with an additional offline phase ('Off-to-On'), and RLPD combined with OPT ('OPT'). The experimental results demonstrate that inte-

grating OPT with RLPD leads to performance improvements, surpassing both the baseline methods. These results indicate that OPT is an effective algorithm capable of enhancing performance when applied to existing state-of-the-art algorithms.

# 6 CONCLUSION

This paper introduced Online Pre-Training for Offline-to-Online RL (OPT), a novel method to improve the fine-tuning of offline pre-trained agents. By incorporating an Online Pre-Training phase to learn a new value function, OPT addresses the limitations of existing offline-to-online RL approaches. Our experiments across multiple D4RL environments demonstrated that OPT consistently outperforms current methods and is versatile across different backbone algorithms. These findings suggest that OPT is a robust and effective solution for enhancing performance in offline-to-online RL. The key contribution of this work lies in the introduction of a new value function for online fine-tuning, in contrast to existing methods that focus on modifying the original value function. However, OPT has its limitations, particularly in the lack of analysis on alternative approaches to training the new value function. Future research could explore different strategies for Online Pre-Training, offering potential improvements to the offline-to-online RL framework.

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

## A  IMPLEMENTATION

### A.1  IMPLEMENTATION DETAILS FOR OPT

We implement OPT on top of each backbone algorithm's code. TD3, RLPD are based on its official implementation[1][2], SPOT, IQL are built upon the CORL (Tarasov et al. 2024) library [3]. The primary modification introduced in our method is the addition of the Online Pre-Training phase. The Online Pre-Training phase is implemented with modifications to OEMA code[4]'s meta-adaptation method, adapted specifically for value function learning. Additionally, the balanced replay (Lee et al. 2022) is implemented using the authors' official implementation[5]. Aside from these changes, no other alterations are made to the original code.

### A.2  ADDITIONAL IMPLEMENTATION DETAILS FOR IQL

The proposed method is applicable to various baseline algorithms. Here, we present the implementation when applied to IQL. IQL trains the state value function and the state action value function as follows:

$$L_V(\mu) = \mathbb{E}_{(s,a)\sim\mathcal{D}}\left[\mathcal{L}_2^\tau(Q_{\bar{\theta}}(s,a) - V_\mu(s))\right]. \tag{5}$$

$$L_Q(\theta) = \mathbb{E}_{(s,a,r,s')\sim\mathcal{D}}\left[(r + \gamma V_\mu(s') - Q_\theta(s,a))^2\right]. \tag{6}$$

where $\mathcal{L}_2^\tau(u) = |\tau - 1(u < 0)|u^2$ and $\tau \in (0,1)$ is the expectile value, and $Q_{\bar{\theta}}$ denotes a target state action value function. Then, using the state action value function and state value function, the policy is trained through Advantage Weighted Regression:

$$L_\pi(\phi) = \mathbb{E}_{(s,a)\sim\mathcal{D}}\left[\exp(\beta(Q_{\bar{\theta}}(s,a) - V_\mu(s)))\log \pi_\phi(a|s)\right] \tag{7}$$

where $\beta \in [0,\infty)$ is an inverse temperature. To apply the proposed method to IQL, we train both a new state action value function ($Q^{\text{on-pt}}$) and state value function ($V^{\text{on-pt}}$) in the Online Pre-Training. The state action value function is trained identically to Eq. 4, while the state value function is trained as follows:

$$\mathcal{L}_V^{\text{pretrain}}(\mu) = \mathcal{L}_V^{\text{off}}(\mu) + \mathcal{L}_V^{\text{on}}(\mu - \alpha\nabla\mathcal{L}_V^{\text{off}}(\mu)). \tag{8}$$

In the online fine-tuning, policy improvement utilizes $\pi^{\text{off}}$, $Q^{\text{off-pt}}$, and $V^{\text{off-pt}}$ from offline pre-training as well as $Q^{\text{on-pt}}$ and $V^{\text{on-pt}}$ from Online Pre-Training. The policy is trained using an advantage weight obtained from $Q^{\text{off-pt}}$ and $V^{\text{off-pt}}$, and a separate weight obtained from $Q^{\text{on-pt}}$ and $V^{\text{on-pt}}$. These two sets of weights are then combined to effectively train the policy:

$$L_\pi(\phi) = \mathbb{E}_{(s,a)\sim\mathcal{D}}\Big[\exp\big(\beta\big(\kappa(Q_{\bar{\theta}}^{\text{off-pt}}(s,a) - V_\mu^{\text{off-pt}}(s)) \tag{9}$$
$$+ (1-\kappa)(Q_{\bar{\psi}}^{\text{on-pt}}(s,a) - V_\nu^{\text{on-pt}}(s)))\big)\log \pi_\phi(a|s)\Big]$$

### A.3  ADDITIONAL IMPLEMENTATION DETAILS FOR RLPD

To verify the effectiveness of the proposed method when applied to ensemble techniques and high replay ratio, we conduct experiments by integrating OPT into RLPD. Since RLPD does not originally include an offline phase, we incorporate an additional offline phase into its implementation. During the offline phase, we follow the RLPD learning method, performing 1M gradient steps. In the online phase, we adhere to OPT by conducting Online Pre-Training for 25k environment steps, followed by 275k environment steps. Given that the original RLPD employs symmetric sampling in the online phase, where half of the samples are sampled from offline data and the other half from online data, we also utilize symmetric sampling instead of balanced replay.

---

[1] https://github.com/sfujim/TD3
[2] https://github.com/ikostrikov/rlpd
[3] https://github.com/tinkoff-ai/CORL
[4] https://github.com/guosyjlu/OEMA
[5] https://github.com/shlee94/Off2OnRL

## A.4 BASELINE IMPLEMENTATION

For comparison with OPT, we re-run all baselines. The results for all baselines are obtained using their official implementations.

- OFF2ON (Lee et al. 2022) : https://github.com/shlee94/Off2OnRL
- OEMA (Guo et al. 2023) : https://github.com/shlee94/Off2OnRL
- PEX (Zhang et al. 2023) : https://github.com/Haichao-Zhang/PEX
- ACA (Yu & Zhang 2023) : https://github.com/ZishunYu/Actor-Critic-Alignment
- FamO2O (Wang et al. 2024) : https://github.com/LeapLabTHU/FamO2O
- Cal-QL (Nakamoto et al. 2024) : https://github.com/nakamotoo/Cal-QL

# B DETAILED DESCRIPTION OF THE ENVIRONMENT AND DATASET.

## B.1 MUJOCO

MuJoCo consists of locomotion tasks and provides datasets of varying quality for each environment. We conduct experiments on the `halfcheetah`, `hopper`, and `walker2d` environments. MuJoCo environment are dense reward setting, and we use the "-v2" versions of the `random`, `medium`, and `medium-replay` datasets for each environment.

## B.2 ANTMAZE

Antmaze involves controlling an ant robot to navigate from the start of the maze to the goal. Antmaze is a sparse reward environment where the agent receives a reward of +1 upon reaching the goal, and 0 otherwise. The maze is composed of three environments: umaze, medium, and large. For the dataset, we use the "-v2" versions of `umaze`, `umaze-diverse`, `medium-play`, `medium-diverse`, `large-play`, and `large-diverse`.

## B.3 ADROIT

Adroit is a set of tasks controlling a hand robot with five fingers. Each environment has a different objectives: in the `pen` environment, the task is twirling a pen; in the `hammer` environment, hammering a nail; in the `door` environment, grabbing a door handle and opening it; and in the `relocate`, locating a ball to goal region. We utilize the "-v1" version of the `cloned` dataset for each environment.

## C  FURTHER EXPERIMENTAL RESULTS

### C.1  ABLATION ON REPLAY BUFFER

The proposed method employs balanced replay (Lee et al. 2022) to enhance the use of online samples during online fine-tuning. Since the balanced replay has proven effective in offline-to-online RL on its own, we conduct experiments to assess its impact and dependency within OPT. To align with OPT's original framework of rapid adaptation through more frequent learning for online samples during online fine-tuning, we test a setup that uniformly samples from online data, a method we refer to as Online Replay (OR).

The results in Table 5 demonstrate that the performance gains of OPT are not solely attributable to the effects of balanced replay. Furthermore, replacing balanced replay with this simpler Online Replay setup still results in significant performance improvements compared with the baseline. These findings indicate that OPT's performance stems not just from balanced replay, but from other strategy that emphasizes learning from online samples, such as Online Replay.

Table 5: Comparing normalized score for OPT with balanced replay and online replay buffer. `Improvement`$(\%)$ refers to the performance gain when compared to the baseline.

| Environment | OPT with BR | OPT with OR |
|---|---|---|
| halfcheetah-r | **89.0** $_{\pm 2.1}$ | 81.4 $_{\pm 6.2}$ |
| hopper-r | 109.5 $_{\pm 3.1}$ | **109.5** $_{\pm 5.1}$ |
| walker2d-r | 88.1 $_{\pm 5.2}$ | **89.7** $_{\pm 21.4}$ |
| halfcheetah-m | **96.6** $_{\pm 1.7}$ | 89.8 $_{\pm 3.0}$ |
| hopper-m | 112.0 $_{\pm 1.3}$ | **111.2** $_{\pm 1.7}$ |
| walker2d-m | **116.1** $_{\pm 4.7}$ | 113.9 $_{\pm 8.2}$ |
| halfcheetah-m-r | **92.2** $_{\pm 1.2}$ | 85.3 $_{\pm 2.7}$ |
| hopper-m-r | **112.7** $_{\pm 1.1}$ | 111.4 $_{\pm 1.8}$ |
| walker2d-m-r | **117.7** $_{\pm 3.5}$ | 109.1 $_{\pm 9.1}$ |
| Total | **933.9** | 901.3 |
| Improvement $(\%)$ | **24.5** $\%$ | 20.2 $\%$ |

### C.2  ABLATION ON $\kappa$

OPT adjusts the weights of $Q^{\text{off-pt}}$ and $Q^{\text{on-pt}}$ during online fine-tuning through the coefficient $\kappa$. To evaluate OPT's sensitivity to $\kappa$ values, we provide additional experimental results. The experiments are conducted in the MuJoCo domain, excluding random datasets, as $\kappa$ is fixed for these cases. To analyze OPT's sensitivity to $\kappa$, we examine an alternative linear scheduling approach where $\kappa$ transitions from 0.2 to 0.8.

Table 6: Results of the ablation study on $\kappa$. All experimental results are measured after 300k steps of online fine-tuning, with 5 random seeds used for each experiment.

| | OPT (medium: $0.1 \to 0.7$ medium-replay: $0.1 \to 0.9$) | Scheduling ($0.2 \to 0.8$) |
|---|---|---|
| ha-m | 96.6 $_{\pm 1.7}$ | 96.2 $_{\pm 1.9}$ |
| ho-m | 112.0 $_{\pm 1.3}$ | 111.3 $_{\pm 0.9}$ |
| wa-m | 116.1 $_{\pm 4.7}$ | 117.7 $_{\pm 1.9}$ |
| ha-m-r | 92.2 $_{\pm 1.2}$ | 92.7 $_{\pm 2.5}$ |
| ho-m-r | 112.7 $_{\pm 1.1}$ | 111.9 $_{\pm 0.7}$ |
| wa-m-r | 117.7 $_{\pm 3.5}$ | 114.2 $_{\pm 6.0}$ |
| Total | 647.3 | 644.0 |

The results indicate that varying the $\kappa$ scheduling has minimal impact on performance. This suggests that the precise values of $\kappa$ are less critical compared to its role in facilitating the transition from $Q^{\text{off-pt}}$ to $Q^{\text{on-pt}}$, which is essential for effective adaptation.

### C.3   FULL RESULTS FOR COMPARISON OF DIFFERENT INITIALIZATION METHODS

In Section 5.1, we conduct an ablation study on Online Pre-Training. Below are the full results of Figure 4.

Table 7: Results of the ablation study on Online Pre-Training. All experimental results are measured after 300k steps of online fine-tuning, with 5 random seeds used for each experiment.

| Environment | OPT | Random Initialization | Pre-trained with $B_{on}$ |
|---|---|---|---|
| halfcheetah-r | $89.0_{\pm2.1}$ | $76.7_{\pm11.9}$ | $68.3_{\pm8.4}$ |
| hopper-r | $109.5_{\pm3.1}$ | $105.8_{\pm7.8}$ | $105.5_{\pm3.2}$ |
| walker2d-r | $88.1_{\pm5.2}$ | $74.8_{\pm10.5}$ | $73.4_{\pm28.9}$ |
| halfcheetah-m | $96.6_{\pm1.7}$ | $89.7_{\pm4.6}$ | $95.7_{\pm0.8}$ |
| hopper-m | $112.0_{\pm1.3}$ | $111.3_{\pm1.9}$ | $109.6_{\pm2.2}$ |
| walker2d-m | $116.1_{\pm4.7}$ | $101.0_{\pm9.2}$ | $114.7_{\pm2.2}$ |
| halfcheetah-m-r | $92.2_{\pm1.2}$ | $84.3_{\pm8.2}$ | $91.0_{\pm1.9}$ |
| hopper-m-r | $112.7_{\pm1.1}$ | $111.7_{\pm1.4}$ | $112.2_{\pm0.7}$ |
| walker2d-m-r | $117.7_{\pm3.5}$ | $104.8_{\pm5.2}$ | $111.6_{\pm8.4}$ |
| Total | 933.9 | 843.1 | 882.0 |

### C.4   FULL RESULTS FOR COMPARISON OF DIFFERENT $N_\tau$

In Section 5.4, we conduct an ablation study on $N_\tau$. Below are the full results of Figure 7.

Table 8: Results of the ablation study on $N_\tau$. All experimental results are measured after 300k steps of online fine-tuning, with 5 random seeds used for each experiment.

| | 6250 | 12500 | 25000 | 50000 | 100000 |
|---|---|---|---|---|---|
| ha-r | $66.4_{\pm9.2}$ | $79.9_{\pm4.8}$ | $89.0_{\pm2.1}$ | $93.6_{\pm2.1}$ | $93.7_{\pm2.8}$ |
| ho-r | $105.8_{\pm1.3}$ | $103.7_{\pm4.7}$ | $109.5_{\pm3.1}$ | $108.4_{\pm5.2}$ | $109.2_{\pm2.0}$ |
| wa-r | $75.9_{\pm8.1}$ | $72.9_{\pm9.5}$ | $88.1_{\pm5.2}$ | $92.4_{\pm8.2}$ | $93.9_{\pm4.0}$ |
| ha-m | $98.6_{\pm1.6}$ | $97.0_{\pm2.5}$ | $96.6_{\pm1.7}$ | $98.4_{\pm3.1}$ | $97.3_{\pm2.1}$ |
| ho-m | $105.8_{\pm2.5}$ | $108.7_{\pm2.0}$ | $112.0_{\pm1.3}$ | $110.2_{\pm1.6}$ | $110.9_{\pm2.0}$ |
| wa-m | $112.8_{\pm7.0}$ | $115.3_{\pm5.1}$ | $116.1_{\pm4.7}$ | $119.5_{\pm4.8}$ | $116.2_{\pm1.5}$ |
| ha-m-r | $89.9_{\pm3.4}$ | $90.4_{\pm1.4}$ | $92.2_{\pm1.2}$ | $92.0_{\pm1.8}$ | $92.6_{\pm2.8}$ |
| ho-m-r | $94.9_{\pm5.5}$ | $111.1_{\pm0.7}$ | $112.7_{\pm1.1}$ | $112.8_{\pm2.2}$ | $111.6_{\pm1.5}$ |
| wa-m-r | $111.2_{\pm2.3}$ | $113.0_{\pm1.8}$ | $117.7_{\pm3.5}$ | $113.2_{\pm2.7}$ | $115.4_{\pm2.8}$ |
| Total | 861.3 | 892.0 | 933.9 | 940.5 | 940.8 |

## D  HYPER-PARAMETERS

In this paper, we present the results of applying OPT to various backbone algorithms. Aside from the hyperparameters listed below, all other hyperparameters are adopted directly from the backbone algorithms. In our proposed Online Pre-Training, we set $N_\tau$ to 25000 and $N_{\text{pretrain}}$ to 50000 for all environments. Additionally, for the MuJoCo domain, we use TD3 with a UTD ratio of 5 as the baseline. In the Adroit domain, we use SPOT as the baseline, trained with layer normalization (Ba 2016) applied to both the actor and critic networks.

As mentioned in Section 3.2, we use the parameter $\kappa$ to assign higher weight to $Q^{\text{off-pt}}$ during the early stages of online fine-tuning, gradually shifting to give higher weight to $Q^{\text{on-pt}}$ as training progresses. We control $\kappa$ through linear scheduling. Table 9 outlines the $\kappa$ scheduling for each environment, where $\kappa_{init}$ represents the initial value of $\kappa$ at the start of online phase, $T_{decay}$ specifies the number of timesteps over which $\kappa$ increases, and $\kappa_{end}$ indicates the final value of $\kappa$ after increase. Notably, in the MuJoCo random environment, as demonstrated in Section 5.2 the value function pre-trained offline exhibits significant bias, so it is not utilized during online fine-tuning.

Table 9: $\kappa$ scheduling method for each environment.

| Environment | $\kappa_{init}$ | $T_{decay}$ | $\kappa_{end}$ |
|---|---|---|---|
| halfcheetah-r | 1 | - | 1 |
| hopper-r | 1 | - | 1 |
| walker2d-r | 1 | - | 1 |
| halfcheetah-m | 0.1 | 150000 | 0.7 |
| hopper-m | 0.1 | 150000 | 0.7 |
| walker2d-m | 0.1 | 150000 | 0.7 |
| halfcheetah-m-r | 0.1 | 150000 | 0.9 |
| hopper-m-r | 0.1 | 150000 | 0.9 |
| walker2d-m-r | 0.1 | 150000 | 0.9 |
| umaze | 0.1 | 100000 | 0.9 |
| umaze-diverse | 0.1 | 100000 | 0.9 |
| medium-play | 0.1 | 100000 | 0.9 |
| medium-diverse | 0.1 | 100000 | 0.9 |
| large-play | 0.1 | 100000 | 0.9 |
| large-diverse | 0.1 | 200000 | 0.9 |
| pen-cloned | 0.1 | 250000 | 0.9 |
| hammer-cloned | 0.1 | 250000 | 0.9 |
| door-cloned | 0.1 | 250000 | 0.9 |
| relocate-cloned | 0.1 | 250000 | 0.9 |

# E    COMPARISON WITH BACKBONE ALGORITHM

Our proposed OPT is an algorithm applicable to value-based backbone algorithms. To evaluate the performance improvement achieved by applying OPT, we compare it against the performance of the backbone algorithms. Table 10 presents the performance of OPT when TD3, SPOT, and IQL are used as the backbone algorithms across all environments. When based on TD3 and SPOT, we observe an average performance improvement of 30%. Additionally, when based on IQL, we observe an average performance improvement of 25%.

Table 10: Average normalized final evaluation score for each environment on the D4RL benchmark. We denote the backbone algorithm as `Vanilla` and the result of the algorithm integrated with OPT as `Ours`. All results are reported as the mean and standard deviation across five random seeds.

| Environment | TD3 | | IQL | |
|---|---|---|---|---|
| | Vanilla | Ours | Vanilla | Ours |
| halfcheetah-r | **96.9** $\pm4.9$ | 89.0 $\pm2.1$ | 33.3 $\pm2.5$ | **45.2** $\pm5.3$ |
| hopper-r | 84.4 $\pm30.1$ | **109.5** $\pm3.1$ | 10.6 $\pm1.5$ | **15.2** $\pm6.7$ |
| walker2d-r | 0.1 $\pm0.0$ | **88.1** $\pm5.2$ | 7.5 $\pm1.6$ | **11.0** $\pm4.6$ |
| halfcheetah-m | 96.1 $\pm1.8$ | **96.6** $\pm1.7$ | 50.2 $\pm0.2$ | **55.5** $\pm0.5$ |
| hopper-m | 84.5 $\pm30.3$ | **112.0** $\pm1.3$ | 61.8 $\pm4.9$ | **94.2** $\pm10.4$ |
| walker2d-m | 102.0 $\pm8.0$ | **116.1** $\pm4.7$ | 86.6 $\pm3.0$ | **91.4** $\pm1.2$ |
| halfcheetah-m-r | 87.5 $\pm1.5$ | **92.2** $\pm1.2$ | 46.2 $\pm0.4$ | **50.2** $\pm1.8$ |
| hopper-m-r | 90.9 $\pm25.4$ | **112.7** $\pm1.1$ | **95.5** $\pm10.2$ | 89.8 $\pm25.5$ |
| walker2d-m-r | 107.7 $\pm7.4$ | **117.7** $\pm3.5$ | 90.3 $\pm6.3$ | **106.0** $\pm2.6$ |
| MuJoCo total | 747.2 | **933.9** (+24.9%) | 482.0 | **558.5** (+15.8%) |
| | SPOT | | IQL | |
| | Vanilla | Ours | Vanilla | Ours |
| umaze | 98.4 $\pm1.8$ | **99.8** $\pm0.4$ | 90.4 $\pm5.3$ | **92.8** $\pm2.9$ |
| umaze-diverse | 55.2 $\pm32.7$ | **97.4** $\pm0.4$ | 30.4 $\pm17.6$ | **90.4** $\pm9.4$ |
| medium-play | 91.2 $\pm3.8$ | **98.2** $\pm1.3$ | 83.2 $\pm4.7$ | **88.4** $\pm1.2$ |
| medium-diverse | 91.6 $\pm3.4$ | **98.4** $\pm1.3$ | 83.2 $\pm2.3$ | **89.6** $\pm2.7$ |
| large-play | 60.4 $\pm21.4$ | **78.2** $\pm4.4$ | 53.0 $\pm6.3$ | **65.8** $\pm2.4$ |
| large-diverse | 69.4 $\pm23.7$ | **90.6** $\pm3.7$ | 51.8 $\pm4.9$ | **64.6** $\pm7.1$ |
| Antmaze total | 466.2 | **562.6** (+20.6%) | 392.0 | **491.6** (+25.4%) |
| pen-cloned | 117.1 $\pm13.4$ | **130.3** $\pm6.8$ | 90.7 $\pm9.4$ | **100.3** $\pm6.0$ |
| hammer-cloned | 90.2 $\pm23.2$ | **120.1** $\pm4.2$ | 14.8 $\pm6.9$ | **23.7** $\pm18.0$ |
| door-cloned | 0.05 $\pm0.06$ | **50.4** $\pm29.2$ | 7.6 $\pm3.4$ | **26.7** $\pm9.3$ |
| relocate-cloned | -0.19 $\pm0.04$ | **-0.11** $\pm0.06$ | 0.09 $\pm0.03$ | **0.83** $\pm0.78$ |
| Adroit total | 207.2 | **300.6** (+45.0 %) | 113.1 | **151.5** (+33.9%) |

# F  FULL RESULTS OF RLPD

Table 11: Average normalized final evaluation score for each environment on the D4RL benchmark. We denote the vanilla algorithm as `Vanilla`, the baseline algorithm within the offline-to-online RL framework as `Off-to-on`, and the result of the algorithm integrated with OPT as `Ours`. All results are reported as the mean and standard deviation across five random seeds.

| Environment | RLPD | | |
|:---:|:---:|:---:|:---:|
| | Vanilla | Off-to-on | Ours |
| halfcheetah-r | $91.5_{\pm 2.5}$ | $\mathbf{96.1}_{\pm 5.2}$ | $90.7_{\pm 2.2}$ |
| hopper-r | $90.2_{\pm 19.1}$ | $95.7_{\pm 18.4}$ | $\mathbf{103.5}_{\pm 3.6}$ |
| walker2d-r | $\mathbf{87.7}_{\pm 14.1}$ | $74.3_{\pm 13.9}$ | $79.2_{\pm 10.0}$ |
| halfcheetah-m | $95.5_{\pm 1.5}$ | $96.6_{\pm 0.9}$ | $\mathbf{96.7}_{\pm 1.4}$ |
| hopper-m | $91.4_{\pm 27.8}$ | $93.6_{\pm 13.9}$ | $\mathbf{106.9}_{\pm 1.5}$ |
| walker2d-m | $121.6_{\pm 2.3}$ | $\mathbf{124.1}_{\pm 2.4}$ | $122.8_{\pm 3.0}$ |
| halfcheetah-m-r | $90.1_{\pm 1.3}$ | $90.0_{\pm 1.4}$ | $\mathbf{91.6}_{\pm 2.1}$ |
| hopper-m-r | $78.9_{\pm 24.5}$ | $94.7_{\pm 26.8}$ | $\mathbf{107.4}_{\pm 1.9}$ |
| walker2d-m-r | $119.0_{\pm 2.2}$ | $\mathbf{122.5}_{\pm 2.7}$ | $120.9_{\pm 2.3}$ |
| MuJoCo total | 866.0 | 887.6 | **918.7** |
| umaze | $99.4_{\pm 0.8}$ | $\mathbf{99.8}_{\pm 0.4}$ | $99.6_{\pm 0.5}$ |
| umaze-diverse | $98.0_{\pm 1.1}$ | $\mathbf{99.2}_{\pm 1.0}$ | $99.0_{\pm 0.6}$ |
| medium-play | $97.6_{\pm 1.4}$ | $97.4_{\pm 1.4}$ | $\mathbf{99.6}_{\pm 0.6}$ |
| medium-diverse | $97.6_{\pm 1.9}$ | $98.6_{\pm 1.4}$ | $\mathbf{99.2}_{\pm 0.4}$ |
| large-play | $\mathbf{93.6}_{\pm 2.4}$ | $93.0_{\pm 2.5}$ | $92.2_{\pm 3.9}$ |
| large-diverse | $92.8_{\pm 3.2}$ | $90.4_{\pm 3.9}$ | $\mathbf{94.8}_{\pm 2.2}$ |
| Antmaze total | 579.0 | 578.4 | **584.4** |
| pen-cloned | $154.8_{\pm 11.6}$ | $148.5_{\pm 15.2}$ | $\mathbf{155.5}_{\pm 11.0}$ |
| hammer-cloned | $139.7_{\pm 5.6}$ | $141.4_{\pm 1.0}$ | $\mathbf{142.1}_{\pm 1.2}$ |
| door-cloned | $110.8_{\pm 6.1}$ | $114.6_{\pm 1.3}$ | $\mathbf{115.7}_{\pm 1.4}$ |
| relocate-cloned | $4.8_{\pm 7.1}$ | $0.11_{\pm 0.2}$ | $\mathbf{10.0}_{\pm 6.4}$ |
| Adroit total | 410.1 | 404.6 | **423.3** |

# G   LEARNING CURVES

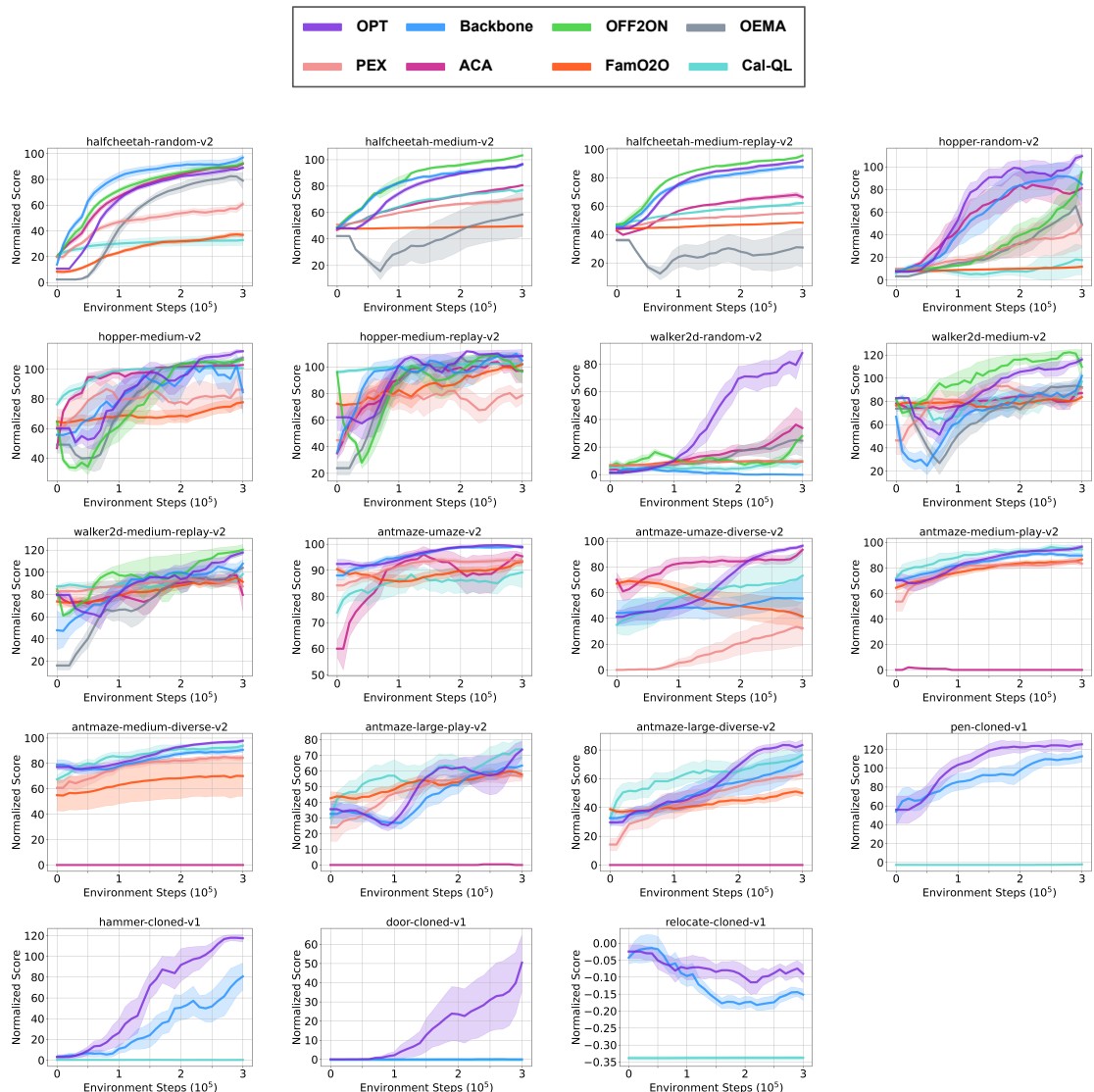

Figure 8: Learning curve of online phase over 300k steps. The solid line represents the mean performance, while the shaded region depicts the standard deviation across five random seeds.

