# OpenReview forum: "Online Pre-Training for Offline-to-Online Reinforcement Learning"
_ICLR.cc/2025/Conference — Submitted to ICLR 2025_

### Official Review · Reviewer_kRi2 · 2024-10-23

**Soundness:** 3
**Presentation:** 3
**Contribution:** 3
**Rating:** 6
**Confidence:** 4

**Summary:**

This paper studies offline-to-online reinforcement learning and finds that directly fine-tuning offline pretrained Q-value functions typically does not lead to good online fine-tuning performance. Therefore, this paper proposes an online pretraining phase for offline-to-online RL, which aims to enable a smoother transition and better fine-tuning results.

**Strengths:**

(1)This paper addresses an important problem, as direct offline-to-online transitions often do not work well. This work can be seen as a method for transitioning offline pretrained Q-values (which are biased due to the behavior cloning term) to adapt more effectively to the level of online values (training from scratch).

(2) The experiments and ablation studies are comprehensive and demonstrate strong results.

**Weaknesses:**

(1)The motivation to address inaccurate value estimation lacks analysis. The current discussions about inaccurate value estimation are derived from other works. I believe the authors should provide deeper insights into why inaccurate value estimation hinders online learning. Is it truly the inaccurate value estimation that causes problems for offline-to-online learning, or is it overfitting to the offline dataset? In some cases, inaccurate value estimation is not necessarily an issue. For example, using pessimistic estimates online does not prevent further improvement but instead helps stabilize value function learning and could lead to good performance [1] [2].

(2)I have some concerns about the sample efficiency of OPT. OPT could be considered a method for transitioning offline-learned value functions, which are based on behavior cloning and temporal difference learning (biased due to the behavior cloning term), to an online level. From the learning curves, it appears that OPT requires completing this adaptation before it starts showing improvements. While I understand that this adaptation inevitably requires some environment interactions, I wonder if this could be improved. For instance, during the pretraining phase, it seems strange that OPT only allows the offline pretrained policy to collect data. Since you are adapting Q with online pretraining (Q on-pt), why not allow a new policy to evolve based on Q on-pt and have it collect data as well? This way, Q on-pt could more easily calibrate its value estimation to the online level based on its own experiences. This might reduce the need to rely on offline data during the online pretraining phase, as trial-and-error could mitigate overestimation and enable further improvement beyond Q off during the transition phase.

(3)I am also concerned about the complexity of OPT. OPT, in its current form, combines different techniques: MEGA for online pretraining and balanced replay for online fine-tuning. This makes OPT seem overly complicated, in my view. For example, if OPT did not use MEGA, would it still work? If it works, then why use MEGA? If not, then why is MEGA necessary? Similarly, why use balanced replay? After the online pretraining phase, shouldn’t Q on-pt already capture the offline pretrained policy’s knowledge by training on B-on, then why still using the offline datasets is important during online phase? Also you are comparing to MEGA and balanced replay in experiments, but you are combining their strategies in some way, and it is not suprising that OPT could be better than MEGA and balanced replay alone (Off2On has the second highest total score in table 2).

[1]Mark, M. S., Sharma, A., Tajwar, F., Rafailov, R., Levine, S., & Finn, C. Offline RL for Online RL: Decoupled Policy Learning for Mitigating Exploration Bias.
[2]Luo, Y., Ji, T., Sun, F., Zhang, J., Xu, H., & Zhan, X. (2024). Offline-Boosted Actor-Critic: Adaptively Blending Optimal Historical Behaviors in Deep Off-Policy RL. arXiv preprint arXiv:2405.18520.

**Questions:**

(1)Why is Qoff-pt also being updated during online fine-tuning? Is updating Qon-pt not sufficient?

(2)Question about the coefficient k: Does it start from a small value? With a small k, it seems like you are adding another adaptation period for OPT. Why is this necessary? After online pretraining, shouldn't Qon-pt already retain the good state-action pairs from Qoff and adjust their values to the online level? Is this additional period introduced because the initial stage of online learning might cause Qon-pt to unlearn?

(3)I’m curious about the performance of OPT with different numbers of offline training gradient steps. Could you provide some insight into how this affects results?

(4)Could you provide more details on training Qon-pt using only Bon? Does Qon-pt produce highly overestimated Q-values for the state-action pairs in Bon, which hinders online improvement?

(5)For the ablation in Section 5.3: How is Qoff-pt trained during online pretraining? Does it continue to collect and store its experiences in Bon and then update using Equation (3)? Why is it that online pretraining with only Qoff-pt cannot correct the Q-values it learned during the offline phase to the level of purely online learning?

(6)How do the baseline methods use the offline dataset during online learning? They might benefit from using the offline dataset online, and this needs further explanation.

(7) Is the basic idea of OPT somehow connected to PEX?


I look forward to discuss with the authors, and increase my score if the questions are answered thoroughly.

---

> ### Author Response · Authors · 2024-11-24
> **Authors' response to reviewer kRi2**
>
> We sincerely thank the reviewer for their thoughtful feedback and for recognizing the significance of our work and its contributions.
>
> &nbsp;
>
> **W1. Regarding Motivation**
>
> The value function learned by offline RL, trained on a fixed dataset, often provides inaccurate value estimation for out-of-distribution samples [4-1].
> Overfitting to the offline dataset may also contribute to this issue, particularly when the dataset is of low quality [4-2].
> During online fine-tuning, the value function naturally encounters such OOD samples, and these inaccurate estimations can negatively impact policy learning and hinder performance improvement, as illustrated in Figure 1 of our paper.
>
> Recent studies such as [4-3,4-4] address this challenge indirectly by adopting conservative policy learning strategies during the online phase to mitigate the effects of inaccurate value estimation.
> In contrast, OPT is specifically designed to directly tackle this issue of inaccurate value estimation during the transition from offline to online learning, making it a distinct approach with a focused objective.
>
> &nbsp;
>
> **W2. Regarding Sample Efficiency**
>
> We appreciate your alternative suggestion regarding the use of a new policy during Online Pre-Triaining.
>
> The characteristic of the backbone model influences the relatively modest performance of OPT in the early stages of training.
> Since OPT builds upon the pre-trained model, the initial performance reflects the early stage performance of backbone model.
> However, as training progresses, $Q^{\text{on-pt}}$ effectively adapts to the samples generated by the improving policy, resulting in a noticeable performance increase, as shown in our learning curves.
>
> The suggestion to incorporate a new policy during Online Pre-Training is an intriguing alternative.
> However, we utilize $\pi^{\text{off}}$ because it already represents a policy derived from the offline dataset, providing a stable and efficient foundation for subsequent online fine-tuning while avoiding the additional exploratory overhead of training a new policy.
> Exploring the integration of a new policy remains an interesting avenue for future work.
>
> &nbsp;
>
> **W3. Regarding OEMA and Balanced Replay**
>
> We provide a detailed explanation of the components of the proposed algorithm in Common Response C.1, where we also demonstrate the contribution of each component through experimental analysis.
> Furthermore, Common Response C.3 elaborates on the necessity and role of meta adaptation within OPT.
>
> Balanced Replay[4-5] is employed in OPT to ensure that $Q^{\text{on-pt}}$, which is designed to adapt effectively to online samples can fully utilize the online data during fine-tuning.
> To analyze the impact of Balanced Replay on OPT, we conduct experiments using an alternative replay strategy, Online Replay, which exclusively utilizes online samples for training.
> As detailed in Appendix C.1, the results show that even with Online Replay, OPT outperforms existing baselines. This indicates that the success of OPT is not solely reliant on Balanced Replay.
>
> Additionally, since OEMA[4-6] also employs Balanced Replay, the comparison between OPT and OEMA highlights that OPT's performance gains are not attributed to Balanced Replay or OEMA individually but rather to their effective combination within the overall algorithm.
> By integrating these components thoughtfully, OPT achieves strong performance without replying on any single technique as the primary driver of its success.
>
> &nbsp;
>
> **Q1 & Q2. Utilizing $Q^{\text{off-pt}}$ and $Q^{\text{on-pt}}$**
>
> At the beginning of online fine-tuning, since the policy has been trained on the offline dataset, it generates samples similar to those in the offline dataset.
> In this setting, $Q^{\text{off-pt}}$, thoroughly trained on the offline dataset, is expected to support the policy learning.
> Based on this, we continue updating $Q^{\text{off-pt}}$ during online fine-tuning and initially assign a lower $\kappa$ value to prioritize its influence.
>
> However, as the policy improves and its distribution begins to diverge from the offline dataset, we gradually increase the weight of $Q^{\text{on-pt}}$, utilizing the adaptation effectively to the online samples.
> This dynamic balancing leverages the $Q^{\text{off-pt}}$ in the early stages and $Q^{\text{on-pt}}$ in subsequent stages, enabling the method to integrate the advantage of both value functions throughout online fine-tuning.

---

> > ### Author Response · Authors · 2024-11-24
> > **Authors' response to reviewer kRi2**
> >
> > **Q3. Impact of Offline Pre-Training Steps**
> >
> > We evaluate the impact of offline pre-training steps by conducting experiments where the current 1M steps are reduced to 500k. The results, shown in table below, indicate that OPT’s performance remains consistent, with minimal differences observed between the two settings. This suggests that OPT performs reliably across different numbers of offline pre-training steps.
> > ||1M|500k|
> > |:-:|:-:|:-:|
> > |ha-r|89.0|90.1|
> > |ho-r|109.5|110.5|
> > |wa-r|88.1|90.6|
> > |ha-m|96.6|98.3|
> > |ho-m|112.0|109.1|
> > |wa-m|116.1|114.5|
> > |ha-m-r|92.2|91.6|
> > |ho-m-r|112.7|112.3|
> > |wa-m-r|117.7|113.5|
> > &nbsp;
> >
> > **Q4. Regarding Section 5.1**
> >
> > In Section 5.1, we present experimental results where $Q^{\text{on-pt}}$ is trained exclusively on $B_\text{on}$ (referred to as **Pre-trained with $B_{\text{on}}$**).
> > For this setup, the loss function during Online Pre-Training is defined as:
> >
> > $ \\mathcal{L}_{Q^{\\text{on-pt}}}^{\\text{pretrain}}=\\mathbb{E}[(Q^{\text{on-pt}}(s,a)-(r+\gamma Q^{\text{on-pt}}(s',\pi(s'))))^2]  $
> >
> > Since $B_{\text{on}}$ is collected using the fixed policy $\pi^{\text{off}}$, training $Q^{\text{on-pt}}$ solely on $B_{\text{on}}$ can lead to overfitting to this limited and non-diverse dataset.
> > This overfitting prevents $Q^{\text{on-pt}}$ from effectively adapting to the new samples as the policy improves during online fine-tuning.
> > While overestimation for $B_{\text{on}}$ might occur in this process, the primary challenge lies in the overfitting of $Q^{\text{on-pt}}$, which hinders policy learning.
> >
> > These findings underscore the importance of adopting a more adaptive learning strategy, such as the meta adaptation approach employed in OPT, to enable $Q^{\text{on-pt}}$ to accommodate new online samples effectively throughout fine-tuning.
> >
> > &nbsp;
> >
> > **Q5. Regarding Section 5.3**
> >
> > In Section 5.3, we conduct an ablation study to evaluate the impact of introducing a new value function.
> > In the \texttt{w/o $Q^{\text{on-pt}}$} variant, the model follows the same sampling process as OPT, collecting $N_{\tau}$ (25,000) samples using $\pi^\text{off}$ and updating $Q^{\text{off-pt}}$ using Equation (3), with $Q^{\text{on-pt}}$ replaced by $Q^{\text{off-pt}}$.
> >
> > The results show that Online Pre-Training with $Q^{\text{off-pt}}$ alone leads to reduced performance.
> > This decline arises because $Q^{\text{off-pt}}$ is heavily trained on the offline dataset, making it less adaptable to new online samples.
> > This issue becomes particularly pronounced in scenarios involving low-quality datasets, such as the random dataset, where the limitation of $Q^{\text{off-pt}}$ are exacerbated.
> > These findings validate the effectiveness of the proposed method, where introducing a new value function ($Q^{\text{on-pt}}$) enables better adaptation to online samples and enhances overall performance during finetuning.
> >
> > &nbsp;
> >
> > **Q6. Usage of the Offline Dataset During Online Learning**
> >
> > In most offline-to-online RL approaches, both offline and online samples are utilized during online learning.
> > Off2On[4-5] introduces balanced replay, prioritizing online samples while still utilizing offline data, and OEMA[4-6] adopts this directly.
> > Meanwhile, PEX[4-7] and Cal-QL[4-8] balance their training data, simply sampling half from the offline dataset and half from online samples.
> > Similarly, OPT employs balanced replay during online fine-tuning to prioritize online samples while leveraging the offline dataset, ensuring effective adaptation to new data.
> >
> > &nbsp;
> >
> > **Q7. Connection Between OPT and PEX**
> >
> > PEX introduces a new policy during online fine-tuning to balance exploration and exploitation, whereas OPT focuses on value estimation by introducing a new value function.
> >
> > Additionally, PEX utilizes two policies and relies on the value function to decide which policy to deploy, whereas OPT employs two value functions simultaneously during online fine-tuning.
> > This fundamental difference highlights how the two approaches are distinct and not directly connected.
> >
> > &nbsp;
> >
> > [4-1] Kumar, Aviral, et al. "Conservative q-learning for offline reinforcement learning." NeurIPS 2020.
> >
> > [4-2] Kong, Rui, et al. "Efficient and Stable Offline-to-online Reinforcement Learning via Continual Policy Revitalization." IJCAI 2024.
> >
> > [4-3] Mark, Max Sobol, et al. "Offline RL for Online RL: Decoupled Policy Learning for Mitigating Exploration Bias." arxiv 2023.
> >
> > [4-4] Luo, Yu, et al. "Offline-Boosted Actor-Critic: Adaptively Blending Optimal Historical Behaviors in Deep Off-Policy RL." ICML 2024.
> >
> > [4-5] Lee, Seunghyun, et al. "Offline-to-online reinforcement learning via balanced replay and pessimistic q-ensemble." CORL 2022.
> >
> > [4-6] Guo, Siyuan, et al. "Sample efficient offline-to-online reinforcement learning." IEEE 2023.
> >
> > [4-7] Zhang, Haichao, We Xu, and Haonan Yu. "Policy expansion for bridging offline-to-online reinforcement learning." ICLR 2023.
> >
> > [4-8] Nakamoto, Mitsuhiko, et al. "Cal-ql: Calibrated offline rl pre-training for efficient online fine-tuning." NeurIPS 2024.

---

> > > ### Comment · Reviewer_kRi2 · 2024-11-24
> > >
> > > Thank you for taking the time and effort to respond to my questions. Overall, I find the idea behind your approach interesting, but it seems overly complicated. While I believe there is certainly room for improvement, given the limited time for rebuttal, it might be challenging to simplify the algorithm at this stage.
> > >
> > > I have increased my score to 6, but not higher, as I believe the approach could be further improved to be simpler. I hope the authors can make this effort in the future.

---

> > > > ### Author Response · Authors · 2024-11-25
> > > > **Authors' response to reviewer kRi2**
> > > >
> > > > Thank you for your thoughtful feedback and for engaging with our work. While we respect your perspective regarding the complexity of our approach, we believe that each component of the current design serves a specific purpose in addressing key challenges in offline-to-online RL.
> > > >
> > > > That said, we see value in exploring ways to streamline the method in future work to enhance its simplicity. We appreciate your reconsideration of the score and your constructive insights.

---

### Official Review · Reviewer_9ohD · 2024-10-30

**Soundness:** 2
**Presentation:** 2
**Contribution:** 2
**Rating:** 6
**Confidence:** 4

**Summary:**

This paper introduced the Online Pre-Training for Offline-to-Online RL (OPT) to address the issue of inaccurate value estimation in offline pre-trained agents. In particular, OPT introduced a new learning phase, named Online Pre-Training, to learn a new value function using both the offline dataset and new online samples. Later, OPT uses two value functions for online policy fine-tuning.

**Strengths:**

- The writing is clear and the paper is easy to follow.
- The idea of using two different value functions is interesting.
- OPT shows promising results in the experiments.

**Weaknesses:**

- Since OPT needs to learn an extra value function, it increases the computational cost.
- OPT introduces some new hyper-parameters, i.e., $N_\tau$, $\alpha$, $\kappa$ that need further tuning.
- Missing recent baselines, i.e., BOORL (https://github.com/YiqinYang/BOORL), SO2 (https://github.com/opendilab/SO2).
- The proposed method is mostly empirical, there is a lack of theoretical analysis of the proposed method.
- No Figure name & caption & xy label in Appendix G. The legend in the image is very obscure.

**Questions:**

- The paper claims OPT helps to learn a more accruate value function in offline-to-online RL. However, there is no direct proof for this point. Does OPT really learns more accurate value functions compared with other baselines? I would suggest the authers to plot the MSE error of the learned value function with an approximated optimal value function across different tasks for different baselines. We can first train a set of online agents until they can achieve near-optimal performance, and use the mean value function of the set of online agents as an approximation of the optimal value function.
- The paper claims OPT can be used with different offline RL algorithms. My question is if $\pi_{off}$ is quite conservative which mostly takes similar actions as in the offline dataset. Then there will be an exploration issue during the online pre-training stage. Because $\pi_{off}$ avoids to take ODD actions, then the data in the $B_{on}$ is likely to be very close to the data in $B_{off}$. I don't think pre-training on such $B_{on}$ will bring any significant improvement.
- From Appendix G, OPT sometimes performs worse than the other baselines in early stages, i.e., hopper-medium-v2, hopper-medium-replay-v2, walker2d-medium-v2, walker2d-medium-replay-v2, antmaze-umaze-diverse-v2, antmaze-large-play-v2. Can the author explain the reasons?
- What is the wall-clock running time of OPT?
- Which task does Figure 7 use? Is the optimal $N_\tau$ the same for all tasks?

---

> ### Author Response · Authors · 2024-11-24
> **Authors' response to reviewer 9ohD**
>
> We sincerely thank the reviewer for their thoughtful feedback and positive remarks on our work.
> We address the raised points below to provide additional clarity and support for our approach.
>
> &nbsp;
>
> **W1 & Q4. Computational Cost of OPT**
>
> To evaluate computational efficiency, we compare the wall-clock running time of TD3+OPT and TD3 in the walker2d-random-v2 environment, where TD3+OPT achieves the most significant performance gains.
> The comparison results are presented in Figure 1 of our anonymous link (https://sites.google.com/view/iclr2025opt).
>
> For TD3, the wall-clock time is approximately 4000 seconds.
> In contrast, TD3+OPT requires 6000 seconds, primarily due to the additional steps for Online Pre-Training ($N_{\tau}$ and $N_{\text{pretrain}}$ are set to 25,000 and 50,000 steps, respectively) and the updates associated with the addition of the new value function during online fine-tuning.
>
> Although OPT requires additional time compared to TD3, this increase reflects the added effort to train the new value function, which is integral to enhancing adaptability and performance during online fine-tuning.
> The trade-off is balanced, as the additional computational cost results in substantial improvements in overall performance.
>
> &nbsp;
>
> **W2. Regarding Hyper-Parameters**
>
> OPT introduces a small number of hyperparameters to support its components, including Online Pre-Training and the use of two value functions during online fine-tuning. These hyperparameters are chosen to require minimal tuning while supporting the method’s effectiveness:
> - $N_{\tau}$ and $N_{\text{pretrain}}$: Consistently set across all domains, with $N_{\tau}$ analyzed in detail in Section 5.4.
> - $\alpha$: Matches the learning rate of the backbone model, ensuring straightforward configuration.
> - $\kappa$: Balances the contributions of $Q^\text{off-pt}$ and $Q^\text{on-pt}$. An analysis of its impact is provided in Common Response C.3.
>
> Overall, while OPT introduces these hyperparameters, their number is minimal, and they are designed to work effectively with simple configurations.
> Detailed analyses in the paper further support their practical application across diverse settings.

---

> > ### Author Response · Authors · 2024-11-24
> > **Authors' response to reviewer 9ohD**
> >
> > **W3. Comparison with Recent Baselines (BOORL, SO2)**
> >
> > We appreciate the reviewer for pointing out the importance of evaluating our method against recent baselines such as BOORL[3-1] and SO2[3-2].
> > BOORL introduces a Bayesian approach to address the trade-off between optimism and pessimism in offline-to-online RL, providing an insightful framework for tackling this fundamental challenge.
> > Similarly, SO2 integrates advanced techniques into Q-function training to mitigate Q-value estimation issues, offering an effective strategy for improving value learning.
> >
> > We conduct experiments using the official implementations of BOORL and SO2, applying the tuning parameters provided by their original authors to ensure fairness.
> > Additionally. to align with our experimental setup, we extend the evaluation of these baselines to 300k steps.
> > The results are presented in below table:
> > |        | TD3+OPT 300k (ours) | BOORL 200k (paper) | BOORL 200k (reproduced) | BOORL 300k (reproduced) | SO2 100k (paper) | SO2 100k (reproduced) | SO2 300k (reproduced) |
> > |:-:|:-:|:-:|:-:|:-:|:-:|:-:|:-:|
> > |  ha-r  |     89.0     |        97.7        |           97.9          |          101.3          |       95.6       |          99.1         |         130.2         |
> > |  ho-r  |     109.5    |        75.7        |          102.2          |          108.5          |       79.9       |          96.2         |          88.5         |
> > |  wa-r  |     88.1     |        93.6        |           4.3           |           0.1           |       62.9       |          36.9         |          66.8         |
> > |  ha-m  |     96.6     |        98.7        |          100.4          |          102.0          |       98.9       |          98.1         |         130.6         |
> > |  ho-m  |     112.0    |        109.8       |          110.4          |          110.7          |       101.2      |         102.5         |          82.1         |
> > |  wa-m  |     116.1    |        107.7       |          105.3          |          114.5          |       107.6      |         109.1         |         105.7         |
> > | ha-m-r |     92.2     |        91.5        |           88.5          |           91.7          |       89.4       |          98.0         |         113.1         |
> > | ho-m-r |     112.7    |        111.1       |          110.9          |          111.0          |       101.0      |         101.8         |          74.4         |
> > | wa-m-r |     117.7    |        114.4       |          109.6          |          112.3          |       98.2       |          96.8         |          27.1         |
> > |  Total |     933.9    |        900.2       |          829.5          |          852.1          |       834.7      |         838.5         |         818.5         |
> >
> > The results show that both SO2 and BOORL, exhibit competitive performance in specific environments, such as halfcheetah, but they generally fall behind OPT in term of average performance across all tasks.
> > For SO2, while strong performance is observed at 100k steps, significant declines occur when extended to 300k steps.
> > BOORL demonstrates comparable performance to OPT in several environments but struggles with reproducibility in walker2d-random (wa-r).
> >
> > BOORL also incurs significantly higher training time due to its computationally intensive ensemble design.
> > Figure 2 of our anonymous link (https://sites.google.com/view/iclr2025opt) highlights this trade-off, showing that OPT achieves comparable or superior performance while maintaining much lower computational costs.
> >
> > &nbsp;
> >
> > **W4. Regarding Theoretical Analysis**
> >
> > Our work focuses on addressing practical challenges in offline-to-online RL through comprehensive empirical evaluation.
> > By conducting extensive experiments across diverse environments, we aim to provide strong evidence of OPT’s effectiveness and adaptability in practice.
> >
> > While theoretical analysis is not the central focus of this work, we believe the empirical findings sufficiently support the proposed approach and its contributions.
> > Nonetheless, we recognize the value of theoretical analysis and consider it a meaningful direction for future research to further reinforce our empirical findings.
> >
> > &nbsp;
> >
> > **W5. Regarding Figures**
> >
> > We appreciate the reviewer’s feedback regarding the clarity of figures in Appendix G of our paper.
> > To address this, we add proper figure names, captions, and x/y axis labels.
> > Additionally, the legend images have been revised for improved readability.
> > These corrections are included in the updated version of the paper.

---

> > > ### Author Response · Authors · 2024-11-24
> > > **Authors' response to reviewer 9ohD**
> > >
> > > **Q1. Regarding Accuracy of Value Estimation in OPT**
> > >
> > > To evaluate the accuracy of value estimation in OPT, we conduct experiments comparing the final value estimation of TD3 and TD3+OPT against an optimal reference.
> > > Specifically, we train TD3 for 1M online steps to achieve near-optimal performance, using its value function as an approximation of the optimal value.
> > > We then evaluate the MSE between the optimal value and the values estimated by each algorithm at 300k steps.
> > >
> > > For this evaluation, we compute value estimates for fixed 10 state-action pairs, where each pair consists of a random initial state and the corresponding optimal action.
> > > The results, in table below, present the MSE for TD3 and TD3+OPT.
> > >
> > > |        |                TD3                | TD3+OPT |
> > > |:-:|:-:|:-:|
> > > |        | MSE ($\downarrow$) |MSE ($\downarrow$)|
> > > |  ha-r  |               433.5               |  2167.4 |
> > > |  ho-r  |                83.2               |   6.6   |
> > > |  wa-r  |              83010.4              |  288.8  |
> > > |  ha-m  |                47.0               |   42.7  |
> > > |  ho-m  |                48.2               |   0.8   |
> > > |  wa-m  |               287.8               |   31.1  |
> > > | ha-m-r |               553.9               |  118.5  |
> > > | ho-m-r |                17.4               |   0.4   |
> > > | wa-m-r |                12.4               |   0.9   |
> > >
> > > The comparison between TD3+OPT and TD3 reveals that TD3+OPT is closer to the optimal value than TD3 in the final stages across most environments.
> > > This improvement is driven by $Q^{\text{on-pt}}$ in TD3+OPT, which adapts more effectively to online samples during fine-tuning.
> > > These results demonstrate that OPT's framework fosters precise value estimation, contributing to its strong overall performance.
> > >
> > > &nbsp;
> > >
> > > **Q2. Regarding Conservative Policies in Online Pre-Training**
> > >
> > > As discussed in Common Response C.3, meta adaptation in Online Pre-Training leverages the offline dataset to enable $Q^{\text{on-pt}}$ to adapt effectively to current policy.
> > > Even if $\pi^{\text{off}}$ is conservatively trained and its trajectory forms a subset of $B_{\text{off}}$, meta adaptation ensures that $Q^{\text{on-pt}}$ adapts specifically to the current policy. This approach is fundamentally different from $Q^{\text{off-pt}}$, which is trained uniformly across $B_{\text{off}}$.
> > >
> > > A similar approach is effectively utilized in OEMA[3-3], where meta adaptation leverages the entire replay buffer alongside a subset of it, referred to as the recent replay buffer, to enhance adaptability to the current policy.
> > > This demonstrates the viability of such an approach and further supports the rationale behind our method.
> > >
> > > &nbsp;
> > >
> > > **Q3. Regarding Early-Stage Performance of OPT**
> > >
> > > OPT's early-stage performance in some environments is influenced by the characteristics of the backbone model it builds upon.
> > > In environments where the backbone model exhibits lower early-stage improvements, this may be reflected in OPT's initial performance as well.
> > >
> > > That said, as training progresses, $Q^{\text{on-pt}}$ effectively adapts to online samples, driving substantial performance improvements in the subsequent stage and showcasing the adaptability of OPT's design.
> > >
> > > &nbsp;
> > >
> > > **Q5. Regarding Figure 7**
> > >
> > > The experiments presented in Figure 7 of our paper are conducted for all 9 environments (3 tasks with 3 datasets each) in the MuJoCo domain, and we include the full results for all $N_{\tau}$ values across tasks in Table 8 (Appendix C.4) in the revised version of the paper.
> > >
> > > The results reveal trends consistent with those shown in Figure 7 across all tasks.
> > > While some environments show improved performance as $N_{\tau}$ increases, larger $N_{\tau}$ values reduce the number of online fine-tuning steps within a fixed total environment step budget.
> > > Based on these observations, we selected 25,000 as the optimal value for $N_{\tau}$, as it balances leveraging $B_{\text{on}}$ and allocating sufficient steps for fine-tuning.
> > >
> > > &nbsp;
> > >
> > > [3-1] Hu, Hao, et al. "Bayesian Design Principles for Offline-to-Online Reinforcement Learning." ICML 2024.
> > >
> > > [3-2] Zhang, Yinmin, et al. "A Perspective of Q-value Estimation on Offline-to-Online Reinforcement Learning." AAAI 2024.
> > >
> > > [3-3] Guo, Siyuan, et al. "Sample efficient offline-to-online reinforcement learning." IEEE 2023.

---

> > > > ### Comment · Reviewer_9ohD · 2024-11-26
> > > >
> > > > Many thanks to the authors for the explanations and experimental results that resolved most of my concerns. I would like to raise my score to 6. I didn't  raise to a higher score because I still find the current method to be a little bit over complicated.

---

> > > > > ### Author Response · Authors · 2024-11-28
> > > > > **Authors' response to reviewer 9ohD**
> > > > >
> > > > > Thank you for your thoughtful feedback and for engaging with our work. While we respect your perspective regarding the complexity of our approach, we believe that each component of the current design serves a specific purpose in addressing key challenges in offline-to-online RL.
> > > > >
> > > > > That said, we see value in exploring ways to streamline the method in future work to enhance its simplicity. We appreciate your reconsideration of the score and your constructive insights.

---

### Official Review · Reviewer_pZqg · 2024-10-31

**Soundness:** 4
**Presentation:** 3
**Contribution:** 3
**Rating:** 6
**Confidence:** 4

**Summary:**

The paper proposes a new improvement technique for offline-to-online RL algorithms to tackle the challenge of slow online fine-tuning from offline pre-training. The key idea is to use separate online pre-training phase to pre-train a new Q-function before fine-tuning the offline-pretrained Q-function and the new Q-function together using regular online RL objective. The online pre-training of the new Q-function leverages a meta-adaptation formulation where the objective is to make the learned new Q-function to be able to adapt to new data encountered during online fine-tuning better. Empirical results show that the proposed can be applied to existing offline RL algorithms like TD3+BC and SPOT to achieve SOTA performance on the D4RL benchmark.

**Strengths:**

- The paper is easy to follow and the empirical evaluations seem thorough and comprehensive.
- Empirical results show the proposed method is effective in improving offline-to-online RL sample efficiency.

**Weaknesses:**

- The online pre-training phase uses a meta-adaptation strategy from prior work that requires meta gradients. This seems to be the most complicated part of the algorithm design. How necessary is this design? Is it sufficient to simply train the new Q-function on the online data with a frozen pre-trained policy? Is that what the second initialization method in Section 5.1 is? It would be good to clarify so.
- Figure 5 and Appendix G have very blurry figure legends.

**Questions:**

- How sensitive is the scheduled of $\kappa$ (the balance between the value functions) on the performance?

---

> ### Author Response · Authors · 2024-11-24
> **Authors' response to reviewer pZqg**
>
> We sincerely thank the reviewer for their insightful feedback and for highlighting key aspects of our work. Your comments have helped us further clarify our approach, and we provide detailed responses to the raised concerns and questions below.
>
> &nbsp;
>
> **W1. Regarding Meta Adaptation Strategy**
>
> We provide a detailed explanation of meta adaptation and its role in Common Response C.3.
> Meta adaptation is a key component that enables $Q^{\text{on-pt}}$ to adapt effectively to online samples during online fine-tuning.
>
> The second initialization method mentioned in Section 5.1 aligns with the approach described in the question, where the new $Q^{\text{on-pt}}$ is trained solely on online data collected from a frozen pre-trained policy.
> However, as the data is generated by a fixed policy, the experimental results show that this approach leads to overfitting to $B_{\text{on}}$.
> This overfitting hinders $Q^{\text{on-pt}}$'s ability to adapt to the new samples generated as the policy improves, resulting in degraded performance.
>
> As a result, this highlights the necessity of meta adaptation in ensuring effective learning and adaptability of $Q^{\text{on-pt}}$ during online fine-tuning.
>
> &nbsp;
>
> **W2. Regarding Figures**
>
> Thank you for your comment regarding the blurry figure legends in Figure 5 and Appendix G.
> We have revised the figure legends to improve clarity and readability.
> Additionally, we have added detailed captions and included appropriate x/y axis labels.
> These updates have been incorporated into the revised manuscript for your review.
>
> &nbsp;
>
> **Q1. Regarding $\kappa$**
>
> In Common Response C.2, experiments with alternative $\kappa$ scheduling (Appendix C.2, Table 6 of the revised paper) demonstrate that performance is consistent regardless of the specific scheduling values.
> This highlights that the gradual transition from $Q^{\text{off-pt}}$ to $Q^{\text{on-pt}}$ is the key factor for effective adaptation, rather than the precise scheduling details.

---

> > ### Comment · Reviewer_pZqg · 2024-11-26
> >
> > Thanks for addressing my concerns and answering all my questions.
> >
> > I do not expect the authors to run more experiment due to the tight rebuttal timeline, but I think that having a more comprehensive ablation experiments on $\kappa$ schedule could further strengthen the paper in the future. The additional experiment in Appendix C.2 Table 6 definitely helped but the alternative $\kappa$ schedule experimented is still relatively similar to the ones used in the paper.

---

> > > ### Author Response · Authors · 2024-11-28
> > > **Authors' response to reviewer pZqg**
> > >
> > > Thank you for your thoughtful feedback and for engaging with our work.
> > >
> > > We acknowledge that exploring a broader range of scheduling strategies for \kappa could further strengthen the paper. While the alternative schedule presented in Appendix C.2 Table 6 provides some insights, we plan to conduct more comprehensive ablation studies on this aspect in future work and update the paper with the findings.

---

### Official Review · Reviewer_K9Bu · 2024-11-05

**Soundness:** 3
**Presentation:** 3
**Contribution:** 3
**Rating:** 6
**Confidence:** 4

**Summary:**

The authors consider the offline-to-online setting, where offline pretraining is followed by finetuning with online interactions. They propose an algorithm which adds another training phase between the offline and online ones---online pretraining---which makes use of online interactions and the offline dataset to learn a seperate value function. This algorithm is shown to improve performance on standard benchmark environments from D4RL.

**Strengths:**

The proposed algorithm and training procedure is clearly explained and shows impressive results on the benchmarks compared to the baselines.
There is also a variety of additional experiments in Section 5 that nicely round out the paper.

**Weaknesses:**

The main issue for me is the complexity of the algorithm and the source of the benefits.
The algorithm involves many new pieces and some important aspects are not studied at all.
For example, the meta-adapation strategy outlined in lines 198-199 is not explained and its contribution to the overall algorithm is unclear. Why is it necessary to train in such a manner instead of, for example, using  $\mathcal{L}^{on}_{Q^{on-pt}}(\psi)$ as the second term in eq.3?


There is also the $\kappa$ weighting parameter, balancing the offline and online losses, which seems important to the algorithm. Schedules for $\kappa$ are discussed in section 5.2. but there's no further description of how exactly it was chosen. Also, the sensivity to this parameter seems important to discuss.

While the benchmark results are good, it is hard to see how these different algorithmic choices impact it and how each of them contribute.

**Questions:**

- Regarding section 5.2: t-sne visualizations can be heavily influenced by its hyperparamters. It would be better to use a different assesmment of distribution differences which are more reliable.

- In table 1, most of the other baseline algortihms (aside from Off2On) underperform vanilla TD3+BC, is this to be expected?

- In table 4, the confidence intervals sometimes overlap significantly. It could be more suitable to not bold any entry in those cases since the performance of the algorithms are similar.


- In the paragraph starting at line 156, it is mentioned that the online pretraining phase helps adjust the value function before the online phase. It would be good to confirm that this is the case as an ablation. In other words, what happens when you directly use a new value function for the online finetuning phase without the pretraining?


- In TD7 [1], the policy gets updated with batches of collected data with the policy fixed to collect each batch.
Online pretraining sounds like something similar, where the current best policy is used to generate data which will be used to perform a batch of updates. This is only done once instead of repeatedly as in TD7 though. Any thoughts on this comparison?


- In fig2, the fire symbol with the red background gives the impression that there is something negative happening in that module. I see it is supposed to contrast the frozen symbol but perhaps a different choice would be better to avoid confusion. For example, using green instead of red and having a checkmark instead or even no symbol.



[1] "For SALE: State-Action Representation Learning for Deep Reinforcement Learning" Fujimoto et al.

---

> ### Author Response · Authors · 2024-11-24
> **Authors' response to reviewer K9Bu**
>
> We sincerely thank the reviewer for their constructive feedback and for highlighting the strengths of our work.
> We address the raised concerns and questions below to provide further clarification and support for our approach.
>
> &nbsp;
>
> **W1 & W3. Clarification of Algorithm Components**
>
> We explain each component of the proposed method and their respective contribution in the Common Response C.1.
> The explanation of meta adaptation strategy and its role are discussed in Common Response C.3.
>
> Performing TD learning solely on $B_{\text{on}}$ is indeed a potential approach for Online Pre-Training.
> However, since $B_{\text{on}}$ is generated from a fixed policy, it lacks diversity and contains a limited number of samples.
> As as result, $Q^{\text{on-pt}}$ is prone to overfitting, which is evidenced by the results presented in Section 5.1 of our paper.
>
> &nbsp;
>
> **W2. Regarding $\kappa$ Scheduling**
>
> We provide additional analysis and experiments related to $\kappa$ in the Common Response C.2.
> These experiments demonstrate that the specific values of $\kappa$ have a minimal impact on overall performance, as long as the balance shifts gradually over time.
> Further details on the scheduling parameters can be found in Appendix D of our paper.
>
> &nbsp;
>
> **Q1. Regarding of t-SNE Visualizations**
>
> To complement the observations presented in the t-SNE visualizations in Section 5.2, we further evaluate the distributional differences using Maximum Mean Discrepancy (MMD) in the walker2d environment.
>
> | | $ B_\text{off} $ $ \leftrightarrow B_\text{init} $ | $B_\text{init} \leftrightarrow B_\text{final}$ | $ B_\text{off} $  $\leftrightarrow B_\text{final}$ |
> |:---:|:---:|:---:|:---:|
> | random | 39.8 | 45.9 | 70.2 |
> | medium | 0.4 | 1.7 | 2.9 |
> | medium-replay | 2.4 | 3.4 | 9.0 |
>
> The MMD results reveal a clear distributional difference in random dataset, while the medium and medium-replay datasets exhibit relatively closer.
> These findings align closely with the trends observed in the t-SNE visualization, confirming the consistency of our results across different evaluation methods.
>
> &nbsp;
>
> **Q2. Clarification of Algorithm Components**
>
> As detailed in Appendix D, we use TD3 with an update-to-data (UTD) ratio of 5 in the MuJoCo environment as the backbone model for our experiments.
> For a fair comparison, we applied the same UTD ratio to TD3 in Table 1.
> This adjustment appears to result in TD3 outperforming other baseline algorithms.
>
> In the revised paper, we explicitly clarify that TD3 is evaluated with a UTD ratio of 5 in Section 4.1.
>
> &nbsp;
>
> **Q4. Regarding Ablation for Online Pre-Training**
>
> To validate the importance of combining Online Pre-Training with a new value function, we conduct an ablation study in Section 5.1 by directly using a random initialized value function for online fine-tuning, without any pre-training.
> This setting, referred to as **Random Initialization**, evaluates whether adding a new value function alone is sufficient to achieve strong performance.
>
> The experiment results, presented in Figure 7 of our paper, demonstrate that $Q^{\text{on-pt}}$, when initialized randomly and fine-tuned without pre-training, generates random and inaccurate estimates during the early stage of online fine-tuning.
> These poor initial estimates limit policy improvement, resulting in significantly degraded overall performance.
>
> This confirms that the mere addition of a new value function is insufficient. Instead coupling it with Online Pre-Trianing enables $Q^{\text{on-pt}}$ to effectively adapt to online samples, facilitating stable and efficient learning during online fine-tuning.
>
> &nbsp;
>
> **Q3 & Q6. Regarding Table 4 and Figure 2**
>
> In response to the observation regarding the confidence intervals in Table 4, we revise the table to ensure that entries are not bolded in cases where confidence intervals overlap significantly.
>
> Regarding the use of the fire symbol in Figure 2, we understand your concern and remove the symbol to avoid potential confusion, following your suggestion.
>
> &nbsp;
>
> **Q5. Comparison with TD7 [1-1]**
>
> TD7 adopts a similar approach by using a fixed policy to collect batches of data, which are then used for updates.
> This design is indeed comparable to our method.
> However, while TD7 employs policy checkpoints primarily for stability, our approach focuses on enabling $Q^{\text{on-pt}}$ to adapt effectively to online samples.
>
> Additionally, our method applied this strategy only once during the learning process, whereas TD7 continuously relies on this mechanism throughout training. This distinction highlights the differing objectives and applications of the two methods.
>
> &nbsp;
>
> [1-1] Fujimoto, Scott, et al. "For sale: State-action representation learning for deep reinforcement learning." NeurIPS 2024.

---

### Author Response · Authors · 2024-11-24
**Common Response**

We sincerely thank the reviewers for their thoughtful feedback and insightful comments, which provided valuable perspectives to clarify our contributions and improve the manuscript. We deeply appreciate the time and effort invested in reviewing our submission.

&nbsp;

**C.1. OPT: Design and Validation of Core Components**

In response to the reviewers' questions regarding the design and significance of each component in OPT, we provide an integrated explanation of their roles alongside the experimental results that validate their effectiveness.

OPT is designed to address the critical issue of inaccurate value estimation during online fine-tuning, a common limitation in offline-to-online RL. To overcome this, we introduce three interdependent components:

1. Introduction of a New Value Function to Mitigate Value Estimation Bias
- A new value function, $Q^{\text{on-pt}}$, is introduced to address the limitations of $Q^{\text{off-pt}}$.
    - Section 5.3 demonstrates substantial performance degradation when this component is removed, particularly in random dataset where $Q^{\text{off-pt}}$ is highly biased.

2. Online Pre-Training for Effective Adaptation
- We introduce an intermediate Online Pre-Training phase to prepare the new value function for online fine-tuning.
    This phase employs a meta adaptation approach that enables the value function to adapt efficiently to newly encountered online data.
    - Section 5.1 demonstrates that the meta adaptation approach is essential for preparing the value function to generalize effectively to new samples during online fine-tuning.

3. Balancing Value Functions During Online Fine-Tuning
- During online fine-tuning, $Q^{\text{off-pt}}$ and $Q^{\text{on-pt}}$ are balanced using $\kappa$, allowing the model to utilize the strengths of both value functions.
    - Section 5.2 explains the rationale for scheduling $\kappa$ to adjust this balance effectively throughout online fine-tuning.

&nbsp;

**C.2. Regarding $\kappa$ Scheduling**

OPT employs a linear scheduling approach for $\kappa$ to transition the contributions from $Q^{\text{off-pt}}$ to $Q^{\text{on-pt}}$ during online fine-tuning.
To assess the sensitivity of this scheduling, we tested an alternative linear schedule where $\kappa$ transitions from 0.2 to 0.8.
The results, presented in Appendix C.2 (Table 6) of the revised paper, reveal that:
- The alternative scheduling achieves performance comparable to the original OPT, suggesting that the specific values of $\kappa$ are not critical.

These findings emphasize that the key factor for effective adaptation is the gradual transition from $Q^{\text{off-pt}}$ to $Q^{\text{on-pt}}$, rather than the precise scheduling details.
This ensures that OPT can adapt efficiently to online samples, maintaining strong performance through fine-tuning.

&nbsp;

**C.3. Explanation about Meta Adaptation**

We aim to provide a detailed explanation of the role and implementation of meta adaptation in Online Pre-Training.

The primary objective of Online Pre-Training is to enable $Q^{\text{on-pt}}$ to adapt effectively to online samples by leveraging the extensive offline dataset alongside the limited online samples generated by the current policy.
To achieve this, we draw inspiration from OEMA[0-1], which employs meta adaptation to ensure that the policy aligns closely with the most recent policy's sample during online fine-tuning.
This aligns with the objective of our Online Pre-Training, where the focus is on enhancing $Q^{\text{on-pt}}$'s adaptability to online samples.

We adopt the approach proposed by OEMA, which is represented in Equation (3) of our paper.
Equation (3) consists of two terms: the first term facilitates learning from the offline dataset, while the second term serves as an objective to ensure that $Q^{\text{on-pt}}$ adapts to online samples.
By optimizing both terms, $Q^{\text{on-pt}}$ is expected to leverage the offline dataset to align closely with the dynamics of current policy, enabling it to adapt efficiently to online samples during online fine-tuning.

Although the formulation in Equation (3) may initially seem complex, it simply combines two standard temporal difference (TD) learning objectives into a loss function, facilitating the intended adaptation with reasonable training complexity.

&nbsp;

**C.4. Paper Revision**

Revised and newly added parts are listed below, with modifications highlighted in blue. Minor typos and formatting inconsistencies are also corrected:

- Section 3.1: We enhance the explanation of meta adaptation.
- Section 3.2: We revise the sentence to eliminate redundancy.
- Figure 5: Blurry labels corrected.
- Appendix C.2: We add ablation results related to $\kappa$.
- Appendix C.4: We include the full results for Figure 7.
- Figure 8: We improve blurry labels, add x/y axis labels to all graphs, and include captions.

&nbsp;

[0-1] Guo, Siyuan, et al. "Sample efficient offline-to-online reinforcement learning." IEEE 2023.

---

### Meta-Review · Area_Chair_52kw · 2024-12-16

**Metareview:**

This work introduces a new training stage between offline pretraining and online training that trains a new value function with both offline data and online interactions. Extensive empirical studies are provided.

My major concern is the rigor of the empirical study and the presentation of the empirical results. It seems to me, all the numbers in the tables are made bold based on only the mean without taking the std into consideration (even if std was considered, given that only 5 seeds are used, the results are still very uninformative). If we take std into consideration, I do not think the proposed method is significantly better than baselines. For example, in Table 1, the proposed method only beat the baseline Off2On in 3 out of 9 tasks, if taking std into consideration. In Table 2, the proposed method only beat the baseline in 1 out of 6 tasks. Making scores bold based only on mean is very misleading and does not make any sense to me when std is actually accompanied. Similarly, the last row of each table is the summation of means. But without second order information, this number is uninformative. Moreover, Section 5.1 is the most important ablation study of the work. But according to the Table 7 in the appendix, taking std into consideration, I think the proposed component is useful only in 1 out of 9 tasks.

Since this paper is purely empirical and there is major concern in the empirical results, I recommend rejection.

**Additional Comments On Reviewer Discussion:**

The reviews are all borderline accept (6666). After I raised my concerns in AC-reviewer discussion, kRi2 expressed similar concerns about the ablation study and questioned the usefulness of the proposed new training stage, despite that kRi2 did not end up with changing their score. On the other hand, no reviewers are against my concerns (the rest three reviewers did not participate in AC-reviewer discussion). I decided to overrule the reviews and recommend rejection. This entails a discussion with SAC and SAC also voted for rejection.

---

### Decision · Program_Chairs · 2025-01-22

Reject